# Electrochemical ammonia synthesis via nitrate reduction on Fe single atom catalyst

Zhen-Yu Wu [1], Mohammadreza Karamad[2], Xue Yong[3], Qizheng Huang[1], David A. Cullen [4], Peng Zhu [1], Chuan Xia[1], Qunfeng Xiao[5], Mohsen Shakouri [5], Feng-Yang Chen [1], Jung Yoon (Timothy) Kim[1], Yang Xia[1], Kimberly Heck[1], Yongfeng Hu [5], Michael S. Wong [1], Qilin Li[6], Ian Gates [2], Samira Siahrostami [3✉] & Haotian Wang [1,7,8,9✉]

Electrochemically converting nitrate, a widespread water pollutant, back to valuable ammonia is a green and delocalized route for ammonia synthesis, and can be an appealing and supplementary alternative to the Haber-Bosch process. However, as there are other nitrate reduction pathways present, selectively guiding the reaction pathway towards ammonia is currently challenged by the lack of efficient catalysts. Here we report a selective and active nitrate reduction to ammonia on Fe single atom catalyst, with a maximal ammonia Faradaic efficiency of ~ 75% and a yield rate of up to ~ 20,000 $\mu g\,h^{-1}\,mg_{cat.}^{-1}$ (0.46 mmol $h^{-1}\,cm^{-2}$). Our Fe single atom catalyst can effectively prevent the N-N coupling step required for $N_2$ due to the lack of neighboring metal sites, promoting ammonia product selectivity. Density functional theory calculations reveal the reaction mechanisms and the potential limiting steps for nitrate reduction on atomically dispersed Fe sites.

[1] Department of Chemical and Biomolecular Engineering, Rice University, Houston, TX, USA. [2] Department of Chemical and Petroleum Engineering, University of Calgary, Calgary, AB, Canada. [3] Department of Chemistry, University of Calgary, Calgary, AB, Canada. [4] Center for Nanophase Materials Sciences, Oak Ridge National Laboratory, Oak Ridge, TN, USA. [5] Canadian Light Source Inc., University of Saskatchewan, Saskatoon, SK, Canada. [6] Department of Civil and Environmental Engineering, Rice University, Houston, TX, USA. [7] Department of Materials Science and NanoEngineering, Rice University, Houston, TX, USA. [8] Department of Chemistry, Rice University, Houston, TX, USA. [9] Azrieli Global Scholar, Canadian Institute for Advanced Research (CIFAR), Toronto, ON, Canada. ✉email: samira.siahrostami@ucalgary.ca; htwang@rice.edu

Ammonia (NH$_3$) is one of the most fundamental chemical feedstocks in thex world, as it is not only an indispensable chemical for fertilizer, pharmaceutical, dyes, etc., but also considered as an important energy storage medium and carbon-free energy carrier[1–5]. Currently, the industrial-scale NH$_3$ synthesis relies on the century-old Haber–Bosch process, which requires harsh operating conditions including high temperature (400–500 °C) and high pressure (150–300 atm) using heterogeneous iron-based catalysts[6–12]. Due to its enormous annual production and energy-intensive processes, the NH$_3$ synthesis industry accounts for 1–2% of the world's energy supply, and causes ca. 1% of total global energy-related CO$_2$ emissions[5,7,9,10]. As an attractive alternative to the Haber–Bosch process, the electrochemical NH$_3$ synthesis route, with renewable electricity inputs such as solar or wind, has attracted tremendous interests over the past few years[4,5,7,9,10,13–17]. Nitrogen gas (N$_2$) from air was identified as one major nitrogen source for this renewable route via electrochemical nitrogen reduction reaction (NRR); however, due to the extremely stable N≡N triple bond (941 kJ mol$^{-1}$) and its non-polarity, NRR suffers from low selectivity (referring to Faradaic efficiency in this work unless otherwise specified) and activity[5,10,13,18–20]. While exciting progresses in NRR catalyst development have been made, in many cases it is still challenging to firmly attribute the detected NH$_3$ to NRR process rather than contaminations due to the extremely low NH$_3$ production rate (mostly <200 μg h$^{-1}$ mg$_{cat.}$$^{-1}$)[5,10,21]. Therefore, using N$_2$ gas as the N source for electrochemical synthesis of NH$_3$, as promising as it is, still has a long way to go to deliver considerable yields for practical applications.

Nitrate (NO$_3$$^-$) ions as one of the world's most widespread water pollutants become an attractive nitrogen source, alternative to the inert N$_2$, for electrochemical synthesis of NH$_3$ (refs. [22–27]). Nitrate source mainly comes from industrial wastewater, liquid nuclear wastes, livestock excrements, and chemical fertilizers, with a wide range of concentrations up to ca. 2 M[23,28–34]. Using electrochemical methods to remove nitrate contaminants from industrial wastewater has been an important topic in environmental research field, and their targeted product of nitrate reduction is N$_2$ instead of NH$_3$ (refs. [29–31,35]). A variety of metal catalysts (including Ru, Rh, Ir, Pd, Pt, Cu, Ag, and Au) and their alloys have been developed over the years to selectively convert NO$_3$$^-$ to N$_2$, with NH$_3$ as the byproducts[28,32,36]. The development of high-performance electrocatalysts to selectively reduce nitrate wastes into value-added NH$_3$ will open up a different route of nitrate treatment, and impose both economic and environmental impacts on sustainable NH$_3$ synthesis.

As the NO$_3$$^-$ reduction to NH$_3$ involves 8$e^-$ transfers and many possible reaction pathways (NO$_2$, NO$_2$$^-$, NO, N$_2$O, N$_2$, NH$_2$OH, NH$_3$, and NH$_2$NH$_2$)[37–39], an in-depth molecular level understanding of elementary steps can guide the rational design of selective catalysts for NH$_3$. As an important competition, NO$_3$$^-$ reduction to N$_2$ pathway involves a N–N coupling step, where two neighboring active sites are possibly needed such as Rh- or Cu-based metal catalysts[32,40]. By dispersing transition metal (TM) atoms into isolated single atoms embedded in supports, the N–N coupling pathway towards N$_2$ gas could be prevented due to the lack of an active neighboring site. As a result, the selectivity towards NH$_3$ could be promoted. Due to this unique atomic structure and electronic property compared to bulk or nanosized TM catalysts, single-atom catalysts (SACs) have attracted tremendous research interests in catalysis field, presenting untraditional activity and selectivity in many catalytic reactions[41–44]. Nevertheless, TM SACs have never been reported for electrocatalytic NO$_3$$^-$-to-NH$_3$ conversion, to the best of our knowledge. More importantly, the well-defined atomic structure of single atomic sites can serve as a great platform to study

nitrate reaction pathways, which are highly complex and poorly understood.

Inspired by the Fe active sites in both Haber–Bosch catalysts (Fe-based compounds) and nitrogenase enzymes (mainly containing Fe–Mo cofactor)[5,10], here we report excellent activity and selectivity of Fe single atomic sites in reducing NO$_3$$^-$ towards NH$_3$. Deposited on a standard glassy carbon electrode, our Fe SAC delivered a maximal NH$_3$ Faradaic efficiency (FE) of ~75% at −0.66 V vs. reversible hydrogen electrode (RHE), with NH$_3$ partial current density of up to ~100 mA cm$^{-2}$ at −0.85 V. This corresponds to an impressive NH$_3$ yield rate of ~20,000 μg h$^{-1}$ mg$_{cat.}$$^{-1}$. Importantly, the Fe SAC displayed a significantly improved NH$_3$ yield rate than that of Fe nanoparticle catalysts despite much lower Fe contents. We used density functional theory (DFT) calculations to elucidate reaction mechanism for NO$_3$$^-$ reduction to NH$_3$ on Fe single atomic site. In addition, we show that NO* reduction to HNO* and HNO* reduction to N* are the potential limiting steps.

## Results

**Synthesis and characterizations of Fe SAC.** The Fe SAC was synthesized by a TM-assisted carbonization method using SiO$_2$ powers as hard templates[45,46]. The strategy involves mixing precursors including FeCl$_3$, o-phenylenediamine with SiO$_2$ powder, followed by pyrolysis of the mixture, then NaOH and H$_2$SO$_4$ etching and second pyrolysis (Fig. 1a; "Methods"). The low-magnification transmission electron microscopy (TEM) image of Fe SAC indicates an interconnected vesicle-like structure with well-defined pores originating from SiO$_2$ hard templates (Fig. 1b and Supplementary Fig. 1). No nanoparticles can be found on the carbon frameworks. Isolated Fe single atoms dispersed on the porous carbon matrix can be clearly identified as bright dots by the aberration-corrected medium-angle annular dark-field scanning transmission electron microscopy (AC MAADF-STEM) image in Fig. 1c ("Methods"). No Fe clusters or nanoparticles are observed in many different areas of Fe SAC (Supplementary Fig. 2). The Fe metal loading is estimated to be 1.51 wt% based on inductively coupled plasma-optical emission spectroscopy (ICP-OES) analysis. Energy-dispersive X-ray spectroscopy (EDS) mapping analysis confirms the existence of Fe, N, and C elements throughout the porous structure (Fig. 1d). A sophisticated point analysis of electron energy loss spectroscopy (EELS) on a single Fe atomic site, as shown in Fig. 1e, confirms the Fe–N–C coordination environment. Considering the angstrom resolution of the electron probe, the signals in EELS point spectrum comes from the Fe atom and its closest neighboring atoms[47,48], suggesting a high possibility of Fe–N direct coordination in Fe SAC. Other point spectra acquired from different areas confirmed similar coordination environments (Supplementary Fig. 3). The X-ray diffraction (XRD) pattern of the Fe SAC exhibits two distinct characteristic peaks at ca. 26.2° and 43.7°, corresponding to the (002) and (101) planes of graphitic carbon (Fig. 1f). There are no characteristic peaks of Fe-based crystals, demonstrating that no large Fe-based crystalline nanoparticles exist in the catalyst. The graphitic carbon structures are also shown by high-resolution TEM as well as Raman spectroscopy (Supplementary Figs. 4 and 5). We used N$_2$ sorption method to analyze the pore structures of the Fe SAC (Fig. 1g), where a remarkable hysteresis loop of type-IV indicates the presence of highly mesoporous structures in Fe SAC. The mesopore size distribution is centered at 18.3 nm (inset in Fig. 1g), and the Brunauer–Emmett–Teller (BET) surface area and pore volume are 285.8 m$^2$ g$^{-1}$ and 0.80 cm$^3$ g$^{-1}$, respectively (Supplementary Table 1).

We further analyzed the chemical and atomic structure of our Fe SAC using X-ray photoelectron spectroscopy (XPS) and X-ray

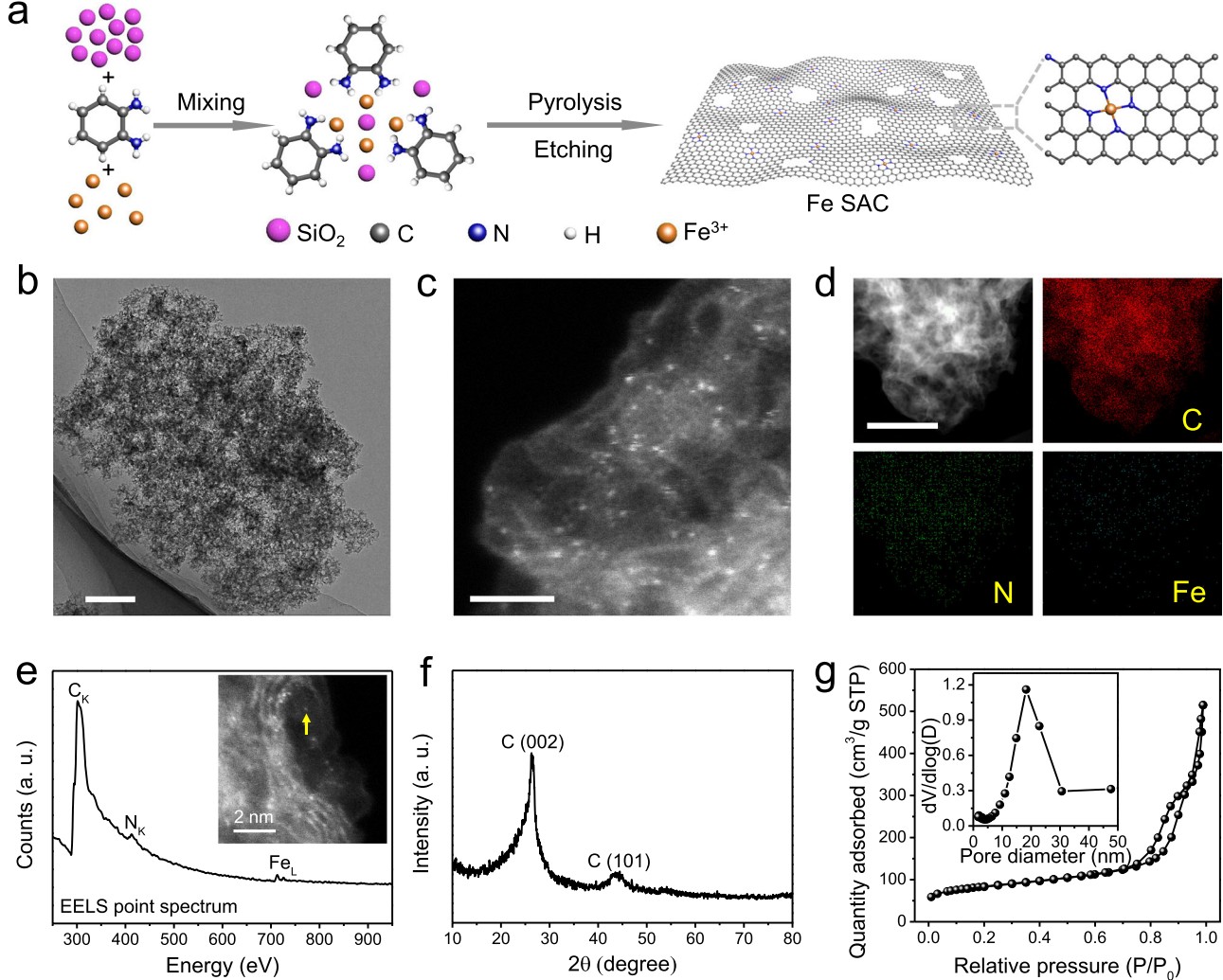

**Fig. 1 Synthesis and characterization of Fe SAC. a** Schematic illustration of the synthesis of Fe SAC. **b** TEM, **c** AC MAADF-STEM, and **d** EDS mapping images of Fe SAC. **e** EELS point spectrum from the Fe atomic site identified by the yellow arrow in the inserted AC MAADF-STEM image of Fe SAC. **f** XRD pattern and **g** $N_2$ adsorption–desorption isotherms of Fe SAC. Inset in **g** is pore-size distribution curve. Scale bars, **b** 200 nm, **c** 2 nm, and **d** 100 nm. Note: a.u. means arbitrary units unless otherwise specified.

absorption spectroscopy (XAS). In XPS results (Fig. 2a and Supplementary Fig. 6a), the high-resolution N 1s spectrum contains four peaks at 398.5, 399.8, 401.0, and 402.6 eV, which are assigned to pyridinic N, pyrrolic N, graphitic N, and oxidized N, respectively[46,49]. No obvious Si 2p XPS signal was found, indicating that $SiO_2$ templates have been completely removed (Supplementary Fig. 6b). The high-resolution Fe 2p spectrum with two relatively weak peaks centered at 711.1 eV (Fe $2p_{3/2}$) and 723.9 eV (Fe $2p_{1/2}$) suggests the positive oxidation states of Fe species in the Fe SAC (Supplementary Fig. 6c)[50]. This is consistent with our XAS analysis (Fig. 2b). The Fe K-edge X-ray absorption near-edge structure (XANES) of Fe SAC presents a near-edge absorption energy between Fe metal foil and $Fe_2O_3$ references, indicating that the oxidation state of Fe single atoms sits between $Fe^0$ and $Fe^{3+}$ (Fig. 2b). The corresponding Fourier-transformed (FT) $k^3$-weighted extended X-ray absorption fine structure (EXAFS) spectrum shows one dominant peak at around 1.6 Å, which can be assigned to the Fe–N coordination at the first shell (Fig. 2c)[49–51]. No Fe–Fe interaction peak at 2.2 Å can be observed, excluding the possibility of any Fe clusters or nanoparticles in our catalyst. These results conclude that the Fe atoms are atomically dispersed in the N-doped carbon (NC) matrix, consistent with our STEM observations. Owing to the

powerful resolutions in both k and R spaces, wavelet transform (WT) of Fe K-edge EXAFS oscillations was employed to further explore the atomic dispersion of Fe in Fe SAC. Only one intensity maximum is observed at ~4.6 Å$^{-1}$ in the WT contour plots, which corresponds to the Fe–N coordination. No intensity maximum belonging to Fe–Fe contribution can be observed, compared with the WT plots of Fe foil and $Fe_2O_3$ (Fig. 2d). To better understand the Fe coordination environment, we also conducted the EXAFS fitting to obtain the structural parameters and extract the quantitative chemical configuration of Fe atoms (Fig. 2e, f). Each Fe atom is coordinated by about 4N atoms in average, and the mean bond length is 1.92 Å (Supplementary Table 2). According to these fitting results, the proposed coordination structure of Fe SAC is Fe–$N_4$, which is shown as the inset in Fig. 2f. The EXAFS fitting results of Fe foil and $Fe_2O_3$ are presented in Supplementary Fig. 7 and Supplementary Table 2. Additionally, only one peak at the $L_3$-edge and no clear multiple structures are found in the Fe L-edge XANES spectrum of Fe SAC, which suggests a unique feature of delocalized Fe 3d electrons of Fe SAC[49]. The itinerant Fe 3d electrons of Fe SAC can be shared by the porphyrin-like structures (as analyzed by Fe K-edge EXAFS fitting) and enhance the electrical conductivity of the catalyst (Fig. 2g)[49]. Other TM SACs including Co and Ni were

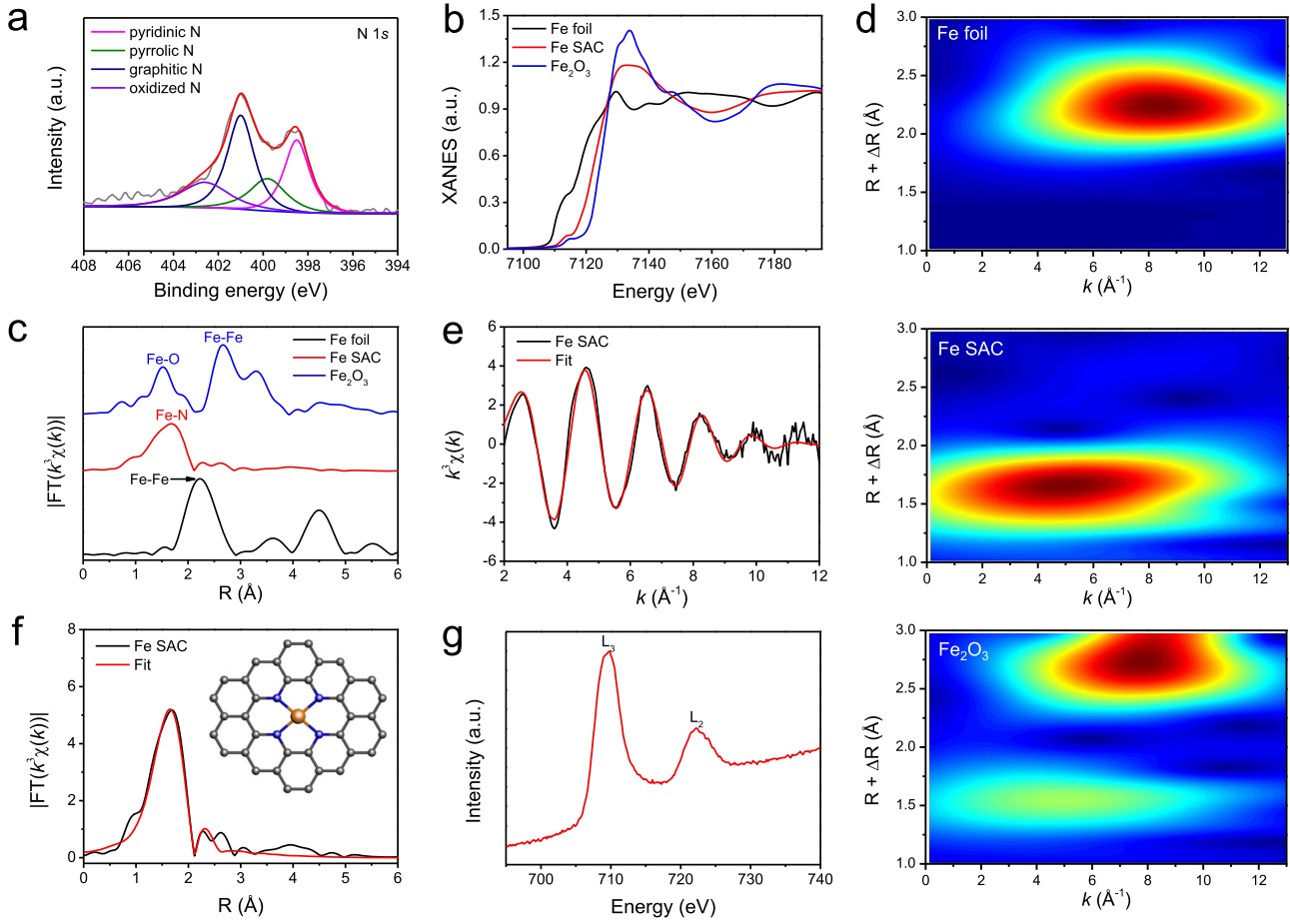

**Fig. 2 Structural analysis of Fe SAC. a** High-resolution N 1$s$ of the Fe SAC. **b** XANES spectra at the Fe K-edge of the Fe SAC, referenced Fe foil and Fe$_2$O$_3$. **c** FT $k^3$-weighted $\chi(k)$-function of the EXAFS spectra at Fe K-edge. **d** WT of the Fe K-edge. Fitting results of the EXAFS spectra of Fe SAC at **e** k-space and **f** R space. Inset: Schematic model of Fe SAC: Fe (yellow), N (blue), and C (gray). **g** XANES spectrum at Fe L-edge of Fe SAC.

also prepared using the same synthesis method, and characterized to confirm their atomic dispersion of TM atoms in NC support (Supplementary Figs. 8–21, Supplementary Tables 1and 2 and Supplementary Note 1).

**Electrocatalytic nitrate reduction performance.** Electrochemical nitrate reduction was conducted in a customized H-cell under ambient conditions. The Fe SAC was deposited onto a mirror-polished glassy carbon electrode with a fixed catalyst mass loading of 0.4 mg cm$^{-2}$. We first performed the linear sweep voltammetry (LSV) in K$_2$SO$_4$ electrolyte with and without KNO$_3$ to study the nitrate reduction catalytic activity of Fe SAC (Fig. 3a). The obviously enhanced current density under the same potential suggests that NO$_3^-$ ions can be effectively reduced by the Fe SAC. Product selectivity was performed in K$_2$SO$_4$/KNO$_3$ electrolyte by holding a certain potential each time for 0.5 h, with generated NH$_3$ products quantified by ultraviolet-visible (UV–Vis) spectrophotometry (Supplementary Fig. 22; see "Methods"). As shown in Fig. 3b, c, our Fe SAC shows high selectivity and superior yield rate for electrocatalytic NO$_3^-$-to-NH$_3$ conversion. At −0.50 V vs. RHE when the reaction starts to onset (an overall current density of 4.3 mA cm$^{-2}$), NH$_3$ product can be readily detected with an FE of 39%, representing a yield rate of 331 µg h$^{-1}$ mg$_{cat.}$$^{-1}$ (Fig. 3b, c). The NH$_3$ selectivity gradually increases to a maximal of ~75% at −0.66 V under an overall current density of 35.3 mA cm$^{-2}$, delivering a yield rate of 5245 µg h$^{-1}$ mg$_{cat.}$$^{-1}$. The NH$_3$ Faradaic efficiency does not

change with time and keeps around 75% during 2 h (Supplementary Fig. 23). A large NH$_3$ partial current density of ~100 mA cm$^{-2}$ is achieved at −0.85 V, corresponding to an impressive yield rate of ~20,000 µg h$^{-1}$ mg$_{cat.}$$^{-1}$. The bare glassy carbon electrode shows a negligible nitrate reduction activity to ammonia (Supplementary Fig. 24). The FE and yield rate of NO$_3^-$-to-NH$_3$ conversion on Fe SAC are orders of magnitude higher than reported N$_2$-to-NH$_3$ conversions[10,21], due to the dramatically different kinetic energy barriers to overcome[39]; more importantly, the ammonia activity per metal active site favorably compare with other nitrate reduction systems which typically used bulk or nanostructured transition metal catalysts (Supplementary Table 3)[22–24,39,52–55]. Different from N$_2$ reduction studies where the concentrations of generated NH$_3$ are typically much lower than $^1$H nuclear magnetic resonance (NMR) detection limit, in our case the generated NH$_3$ has concentrations high enough to be accurately quantified by NMR test, which helps to independently confirm our UV–Vis test. We chose the maximal FE point to be validated by NMR (see "Methods"). As shown in Fig. 3d, three peaks corresponding to $^{14}$NH$_4^+$ are clearly observed in electrolytes after 0.5-h electrolysis under −0.66 V. Based on the averaged NMR peak areas of three independent electrolysis tests and the calibration curve of $^{14}$NH$_4^+$ (Supplementary Fig. 25), we obtained an FE of NH$_3$ at ~76% (Fig. 3b), in good agreement with our UV–Vis spectrophotometry measurements. Additionally, we used NMR to confirm that the NH$_3$ produced actually came from NO$_3^-$ ions using $^{15}$N-labeled NO$_3^-$ (Fig. 3e). Only two peaks of $^{15}$NH$_4^+$

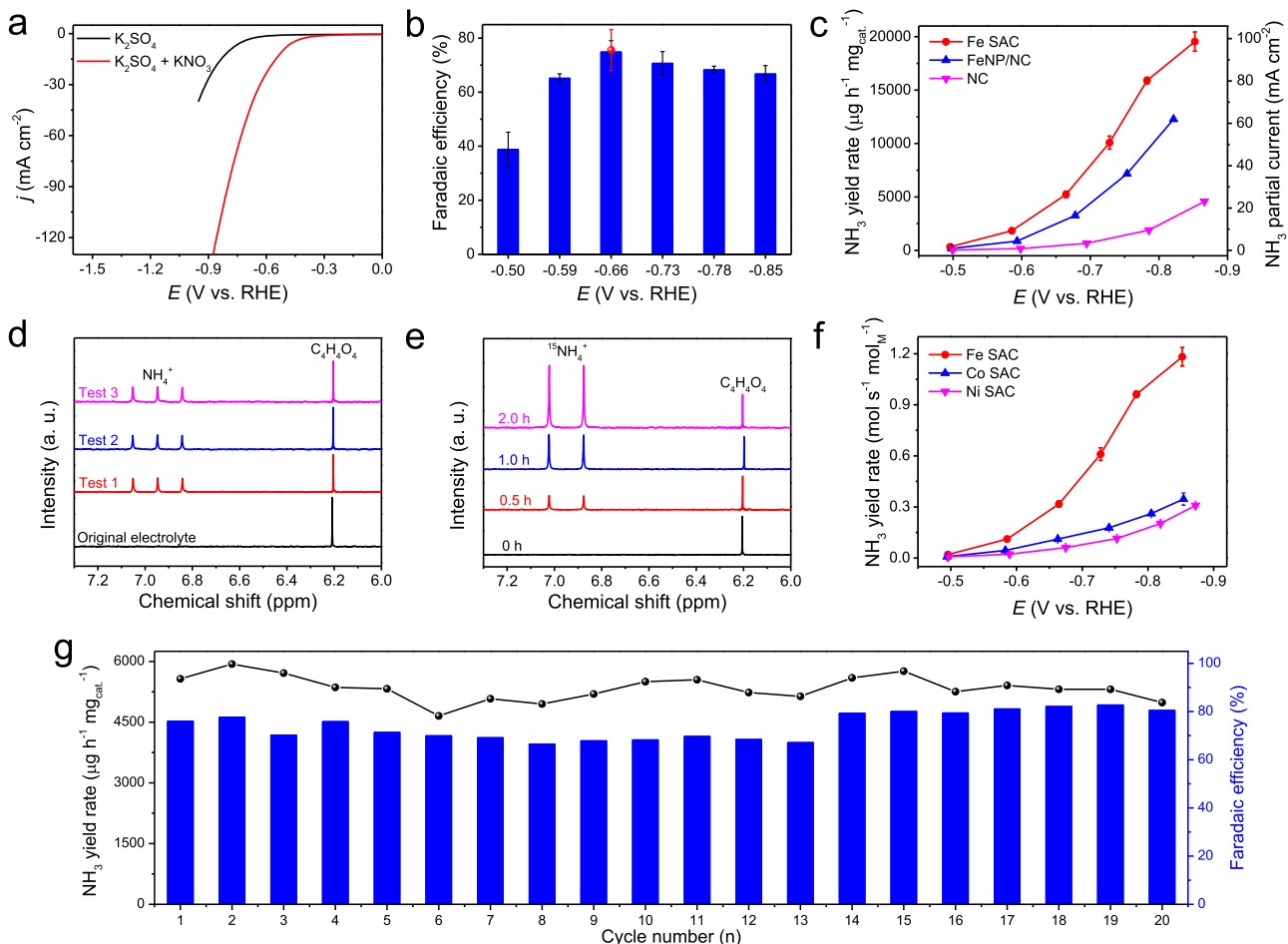

**Fig. 3 Electrocatalytic nitrate reduction performance. a** LSV curves of the Fe SAC in 0.25 M $K_2SO_4$ electrolyte and 0.50 M $KNO_3$/0.10 M $K_2SO_4$ mixed electrolyte. **b** $NH_3$ FE of Fe SAC at each given potential. Red dot is FE estimated by three independent NMR tests. **c** $NH_3$ yield rate and partial current density of Fe SAC, FeNP/NC, and NC. **d** $^1H$ NMR spectra for the electrolytes after three independent nitrate reduction tests at −0.66 V. **e** $^1H$ NMR spectra for electrolytes after $^{15}NO_3^-$ reduction tests at different time using 0.50 M K$^{15}NO_3$/0.10 M $K_2SO_4$ mixed electrolyte. **f** $NH_3$ yield rate of Fe SAC, Co SAC, and Ni SAC based on metal content. **g** The cycling tests of Fe SAC for reduction tests at −0.66 V. Catalyst loading for all of electrocatalytic nitrate reduction tests is 0.4 mg cm$^{-2}$.

appear in $^1H$ NMR spectra with their peak intensity increasing with the electrolysis time, confirming that the $NH_3$ generated is from electrochemical nitrate reduction rather than contaminations. Also, no $NH_3$ could be detected if $KNO_3$ was absent in the electrolyte during the electrolysis (Supplementary Fig. 26).

The main byproduct of nitrate reduction on Fe SAC is $NO_2^-$, the simplest nitrate reduction product, as detected and quantified by UV–Vis (Supplementary Figs. 27 and 28). The FE of $NO_2^-$ starts from as high as 66% at the onset potential, followed by a gradual decease to a minimal of ~9%. This trend correlates to the gradual increase of $NH_3$ selectivity, suggesting that $NO_2^-$ could be an intermediate product and can be further reduced to $NH_3$ under more negative potentials. This hypothesis was further validated by performing $NO_2^-$ reduction on Fe SAC, where more than 90% FE of $NH_3$ and higher production rates can be achieved under the studied potential window (Supplementary Fig. 29). Other possible minor products such as $N_2$ and $H_2$ were further quantified by gas chromatography, with FEs less than 1%. In fact, gas bubbles could hardly be observed on the working electrode during electrolysis until the potential is more negative than −0.73 V.

As various nitrate concentrations exist in different sources, we also evaluated the catalytic performance of Fe SAC at initial $KNO_3$ concentrations ranging from 0.05 to 1.0 M. The maximal FEs of $NO_3^-$-to-$NH_3$ conversion were 74.3, 71.8, and 73.5% in

0.05, 0.1, and 1 M $KNO_3$, respectively, similar to the performance tested in 0.5 M $KNO_3$ solution (Supplementary Fig. 30). This suggests that the $NO_3^-$ concentration has no obvious impacts on Fe SAC's $NH_3$ selectivity. In addition, we observed that the $NH_3$ yield rate was greatly enhanced by increasing the $KNO_3$ concentrations from 0.05 to 0.5 M, but remained nearly unchanged with further increase to 1.0 M. We found that this performance difference was not due to the mass diffusion limit or the concentration of $K^+$ (Supplementary Figs. 31 and 32). There could be a transition of rate-limiting step in the kinetic regime from positive to zero order in nitrate from 0.05 to 1.0 M. For some practical applications, when the nitrate concentrations are much lower or higher in some sources, some strategies could be adopted, such as using mature industrial concentrating processes to concentrate those low-concentration nitrates, and diluting highly concentrated nitrates before conversion, as well as electrochemical cell engineering technology[56,57]. Besides, we found that the presence of NaCl in the electrolyte did not affect the catalytic performance of Fe SAC for nitrate reduction (Supplementary Fig. 33). We also investigated nitrate reduction on Fe SAC at different pH (Supplementary Fig. 34). The FE of $NO_3^-$-to-$NH_3$ conversion in the alkaline solution (pH = 13) is similar to that at neutral pH, with significantly improved overpotentials, while the catalytic activity and selectivity are

significantly lower in acidic solution (pH = 1). In addition, the FE of NO$_3^-$-to-NH$_3$ conversion can be enhanced by further optimizing our catalysis system (Supplementary Fig. 35). An FE of 86% for NO$_3^-$-to-NH$_3$ conversion and NH$_3$ partial current of 60.7 mA cm$^{-2}$ were achieved at −0.21 V in 0.1 M KNO$_3$/1.0 M KOH mixed electrolyte for 2-h electratalysis test (Supplementary Fig. 35 and Supplementary Table 3).

To explore the active sites in our Fe SAC, control experiments in NC and Fe nanoparticles supported on NC (FeNP/NC) were performed to compare with Fe SAC (Supplementary Figs. 36 and 37; see "Methods"). NC support exhibits much lower NH$_3$ activity compared to Fe SAC (Fig. 3c). Although FeNP catalyst shows similar NH$_3$ FE to Fe SAC (Supplementary Fig. 38b), the NH$_3$ yield rate of FeNP/NC was significantly lower than that of Fe SAC (Fig. 3c). Once normalized by metal contents, the NH$_3$ yield rate of Fe SAC per molar Fe is ~20 times higher than that of the NP counterparts, revealing the extraordinary activity on Fe single atomic site (Supplementary Fig. 39 and Supplementary Note 2). The double-layer capacitance (C$_{dl}$) which is proportional to the electrochemical surface area of Fe SAC and FeNP/NC are very close, further demonstrating the intrinsically higher activity of Fe SAC than FeNP/NC (Supplementary Fig. 40). Additionally, we found that the FeNP/NC catalyst was not stable during the nitrate reduction process (−0.87 V for 0.5 h); ~20% of Fe contents were dissolved into the electrolyte solution. Such metal contamination from catalysts is problematic for many applications. As a sharp contrast, no Fe species were detected by ICP-OES in electrolytes after 0.5-h nitrate reduction on Fe SAC under −0.5 V and −0.85 V, suggesting the high stability of Fe single atoms. Also, Fe SAC exhibits much better performance than bulk Fe foil electrode (Supplementary Fig. 41). We also compared the Fe SAC with other TM SACs such as Co and Ni prepared using the same synthesis method. While Co and Ni SACs showed only slightly lower NH$_3$ selectivity, their atomic site activities were around three (Co SAC) and four (Ni SAC) times lower than that of Fe (Fig. 3f, Supplementary 42 and Supplementary Note 3), suggesting the unique activity of Fe atom centers. However, Co and Ni SACs showed much higher activity than NC (Supplementary 43).

The durability of Fe SAC in nitrate reduction was first evaluated by 20 consecutive electrolysis cycles in a H-cell reactor under the best NH$_3$ selectivity reaction condition (Fig. 3g; "Methods"). The NH$_3$ yield rate and FE in each cycle fluctuate but remain stable, suggesting the excellent stability of our catalyst. Importantly, the MAADF-STEM and AC MAADF-STEM images (Supplementary Fig. 44), EELS point spectra (Supplementary Fig. 45) and XAS tests (Supplementary Fig. 46) show that the structure of the Fe SAC is maintained well after the cycling test. Additionally, a 35-h continuous electrolysis was performed in a flow cell reactor under the similar operation current of −35 mA cm$^{-2}$ (see "Methods"), showing negligible changes in working potential or NH$_3$ FE (Supplementary Fig. 47).

**DFT calculations**. We performed DFT calculations to investigate the reaction mechanism and unravel the origin of Fe SAC's high performance in nitrate reduction (Fig. 4; see "Methods"). Based on our characterization results, we used the Fe–N$_4$ motif with Fe atom as the active site in our model. We first investigated different possible reaction pathways for the formation of products such as NH$_3$, NO, N$_2$O, and N$_2$ (Supplementary Fig. 48). Supplementary Fig. 49 only displays pathways that result in NH$_3$ as the main product through nitrate reduction:

$$NO_3^- + 9H^+ + 8e^- \rightarrow NH_3 + 3H_2O \qquad E^0 = 0.88\,V \qquad (1)$$

Nitrate reduction to ammonia is accompanied by nine proton and eight electron transfers. The first step is protonation of NO$_3^-$ which is a solution-mediated proton transfer to form HNO$_3$ and does not require electron transfer. The intermediates and their energy profile across the reaction coordinate are displayed in the free energy diagram in Supplementary Fig. 50. Figure 4a (also the green arrows in Supplementary Fig. 49 and green line in Supplementary Fig. 50) indicate the minimum energy pathway (MEP) for nitrate reduction to NH$_3$ on Fe single atom site. We find that reduction of NO$_2^-$ to NO is downhill in free energy. This finding is in agreement with a previous computational report on Pd surface[58]. Nitrate reduction on polycrystalline and single crystals of transition metals have been studied in the past[34,59–64]. Liu et al.[40] suggested that N* and O* binding energies can be used as descriptors for nitrate reduction performance on TMs. In addition, it has been shown that the main product of nitrate reduction reaction on all transition metals is nitrogen with low selectivity towards ammonia. The latter is due to the dominance of parasitic hydrogen evolution reaction. Moreover, NO* has been suggested as a key intermediate for nitrate reduction on metal surfaces such as Pt where its reduction to HNO* or NOH* is a critical step for production of NH$_4^+$. Our analysis on Fe SAC shows that NO* is a key intermediate for nitrate reduction reaction which is consistent with previous computational reports on Pt and Pd[58,65]. We would like to emphasize that while NO$_2^-$ is confirmed as an intermediate product in the experimental result, our DFT calculations show that the potential limiting steps are the NO* reduction to HNO* and HNO* reduction to N* in agreement with previous computational reports on transition metals such as Pt and Pd[58,65]. Compared to the MEP at U = 0.0 V vs. RHE (green line) in Fig. 4b, a limiting potential of U = −0.30 V (black line) is needed to make all steps downhill in free energy. Although not exactly the same, the calculated limiting potential (−0.30 V) is reasonably comparable with the observed experimental onset potential at ~−0.40 V. The 0.10 V difference can be attributed to the additional kinetic barriers that need to be overcome. We note here that due to the single atom nature of active sites in our catalyst, it is energetically unfavorable to make N–N coupling intermediates or products such as N$_2$O or N$_2$ (Supplementary Fig. 50), which is why we did not observe any N$_2$ products from nitrate reduction on Fe SAC. In addition, our DFT calculations show that the MEP on Fe(110) is different from the one on Fe SAC and the potential limiting step is reduction of NH* to NH$_2$* (Supplementary Fig. 51). The DFT calculated limiting potential on Fe(110) is 0.50 V indicating that Fe(110) exhibits lower catalytic activity than Fe SAC. We also calculated the free energy diagrams for Co and Ni SACs (Supplementary Fig. 52). As it can be seen the potential limiting step is the reduction of NO* to HNO* on both Co and Ni SACs. The calculated limiting potentials for nitrate reduction on Co and Ni, and Fe SACs are 0.42, 0.39, and 0.3 V, respectively, explaining why Fe SAC is more active than Co and Ni SACs. Of note, the potential limiting steps on Co and Ni SAC are highly close and within the range of DFT calculations error, indicating that they have very similar nitrate reduction activity, consistent with our experimental data (Fig. 3f). Combining experimental results and DFT calculations, the high NH$_3$ yield rate or activity of Fe SAC in this study can be attributed to the following two aspects. On the one hand, the Fe SAC has intrinsically high-efficiency active sites, i.e. Fe–N$_4$ centers, which exhibit much lower thermodynamic barriers, evidencing from smaller calculated limiting potentials than that of FeNP of FeNP/NC, Co–N$_4$ of Co SAC, and Ni–N$_4$ of Ni SAC. One the other hand, the optimized electrocatalytic conditions, including the concentration of KNO$_3$, pH of

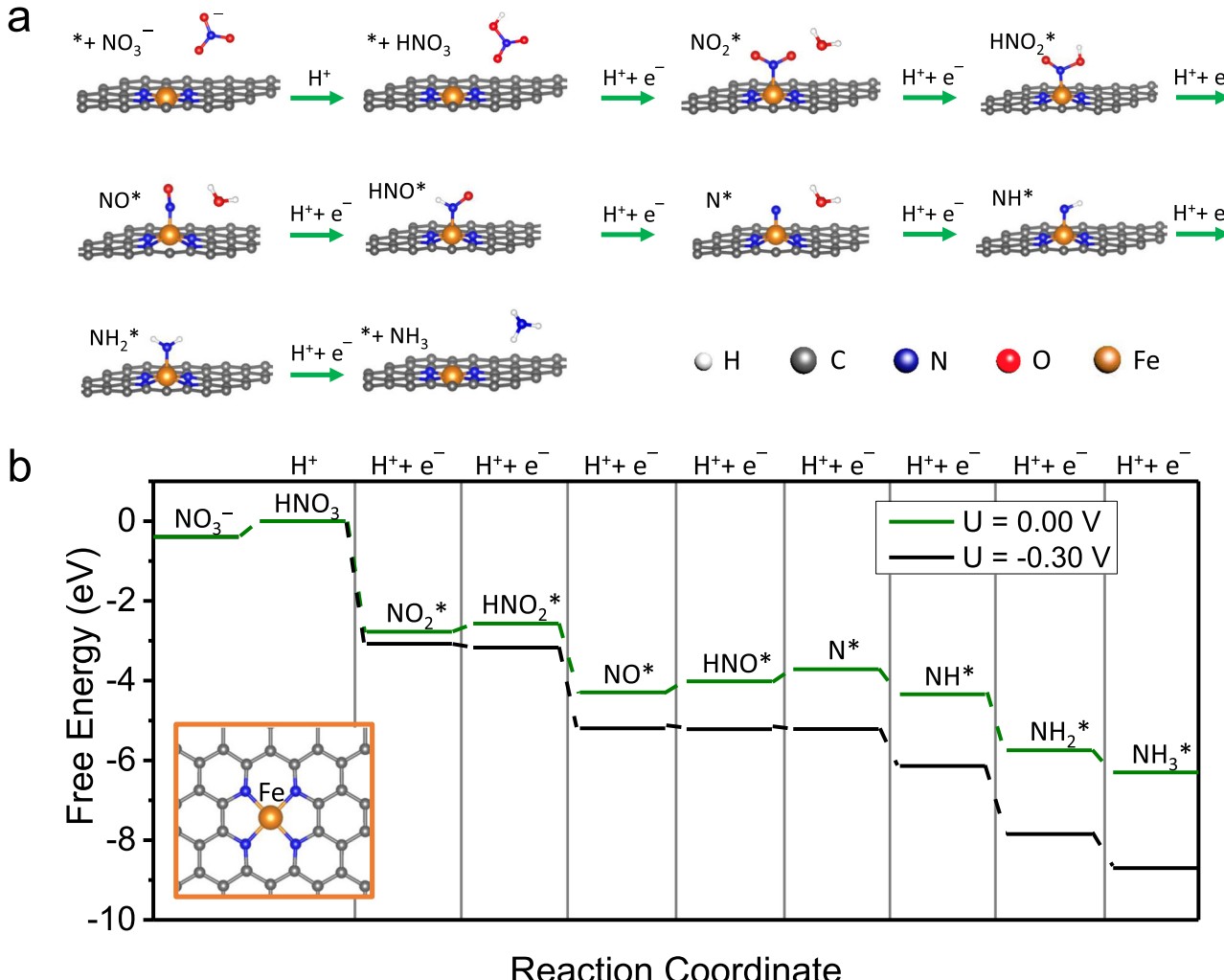

**Fig. 4 DFT calculations. a** The minimum energy pathway that results in NH₃ as the main product. **b** Free energy diagram showing the minimum energy pathway at $U = 0.0$ V vs. RHE (green) and at the calculated limiting potential of $-0.30$ V vs. RHE (black).

electrolyte, and applied potential, also play an important role in high NH₃ yield rate of Fe SAC.

## Discussion

In summary, we have demonstrated Fe SAC as an active and selective electrocatalyst to reduce nitrate to valuable ammonia. Our DFT simulations reveal the reaction pathways and potential limiting steps for nitrate reduction on Fe single atomic site. We believe this nitrate reduction to ammonia route could stimulate a different perspective towards how delocalized ammonia generation could be achieved. Future works should focus on further enhancing the catalytic selectivity, activity, and energy conversion efficiency in nitrate reduction to ammonia, testing the system in real wastewater system, and designing electrochemical reactors for more concentrated ammonia product generated from low-concentration nitrate sources.

## Methods

**Synthesis of Fe SAC.** In a typical synthesis, 2.0 g o-phenylenediamine, 0.58 g FeCl₃, and 2.0 g SiO₂ powder (10–20 nm, Aldrich) were added into 240 mL iso-propyl alcohol and then vigorously stirred for ca. 12 h. After drying the mixture by using a rotary evaporator, the obtained dried powder was subsequently carbonized under flowing Ar for 2 h at 800 °C. Then, the product underwent alkaline (2.0 M NaOH) and acidic (2.0 M H₂SO₄) leaching successively to remove SiO₂ templates and unstable metallic species, respectively. Finally, the Fe SAC was obtained by second heat treatment at the same temperature (i.e. 800 °C) under flowing Ar for another 2 h.

**Synthesis of Co SAC and Ni SAC.** The synthesis processes of Co SAC and Ni SAC are similar to that of Fe SAC, with the only difference being that 0.44 g CoCl₂ and 0.44 g NiCl₂·6H₂O were used for synthesizing Co SAC and Ni SAC, respectively.

**Synthesis of NC catalyst.** For comparison, the NC catalyst was also prepared. Typically, 2 g o-phenylenediamine was firstly dissolved in 30 mL 1.0 M HCl, and then 2.0 g SiO₂ powder was added into the above solution. After stirring for 0.5 h, 24 mL 1.0 M HCl solution containing 6.0 g ammonium peroxydisulfate, i.e., (NH₄)₂S₂O₈, was added dropwise with stirring. The polymerization process was carried out in an ice bath for ca. 24 h. The mixture was dried by using a rotary evaporator, and then carbonized under flowing Ar for 2 h at 800 °C. The SiO₂ templates were removed by 2.0 M NaOH solution. Finally, the NC catalyst was obtained by second heat treatment at the same temperature (800 °C) under flowing Ar for another 2 h.

**Synthesis of FeNP/NC catalyst.** Firstly, 0.528 g FeSO₄·7H₂O and 0.16 g NC were added into 15 mL deionized water and sonicated for 30 min. Then, 10 mL NaBH₄ (containing 0.284 g NaBH₄) aqueous solution was added dropwise into the above solution with vigorous stirring. Then, the mixed solution was stirred for 3 h. The sample was finally obtained by centrifugation collection, thoroughly washing with ethanol and deionized water and drying in an oven. The content of Fe in the FeNP/ NC catalyst was 22.2 wt%, which was determined by ICP-OES analysis.

**Characterization.** TEM observations and EDS elemental mapping were carried out on a Talos F200X transmission electron microscope at an accelerating voltage of 200 kV equipped with an energy-dispersive detector. XPS was performed on an X-ray photoelectron spectrometer (ESCALab MKII) with an excitation source of Mg Kα radiation (1253.6 eV). XRD data were collected on a Rigaku D/Max Ultima II Powder X-ray diffractometer. N₂ adsorption–desorption isotherms were recorded on an ASAP 2020 accelerated surface area and porosimetry instrument

(Micromeritics), equipped with automated surface area, at 77 K using Barrett–Emmett–Teller calculations for the surface area. Raman scattering spectra were obtained by using a Renishaw System 2000 spectrometer using the 514.5 nm line of an $Ar^+$ laser for excitation. Aberration-corrected MAADF-STEM images and EELS point spectra were captured in a Nion UltraSTEM U100 operated at 60 keV and equipped with a Gatan Enfina electron energy loss spectrometer at Oak Ridge National Laboratory. Inductively coupled plasma-atomic emission spectrometry data were recorded on an Optima 7300 DV instrument.

**XAS measurement and data analysis**. XAS spectra at the Fe, Co, and Ni K-edge were measured at the beamline 1W1B station of the Beijing Synchrotron Radiation Facility (BSRF), China. The Fe, Co, and Ni K-edge XANES data were recorded in a fluorescence mode. Fe, Co, and Ni foils and $Fe_2O_3$, $Co_2O_3$, and NiO were used as the references. The storage ring was working at the energy of 2.5 GeV with an average electron current of 250 mA. The hard X-ray was monochromatized with Si (111) double crystals. The acquired EXAFS data were extracted and processed according to the standard procedures using the ATHENA module implemented in the IFEFFIT software packages. The $k^3$-weighted EXAFS spectra were obtained by subtracting the post-edge background from the overall absorption and then normalizing with respect to the edge-jump step. Subsequently, $k^3$-weighted $\chi(k)$ data in the k-space were Fourier transformed to real (R) space using a hanning windows to separate the EXAFS contributions from different coordination shells. To obtain the quantitative structural parameters around central atoms, least-squares curve parameter fitting was performed using the ARTEMIS module of IFEFFIT software packages. The X-ray absorption L-edge spectra of Fe, Co, and Ni were performed at the Catalysis and Surface Science Endstation at the BL11U beamline in the National Synchrotron Radiation Laboratory (NSRL) in Hefei, China.

**Electrocatalytic nitrate reduction**. The electrochemical measurements were carried out in a customized H-type glass cell separated by Nafion 117 membrane (Fuel Cell Store) at room temperature. A BioLogic VMP3 workstation was used to record the electrochemical response. In a typical three-electrode system, a saturated calomel electrode (SCE, CH Instruments) and a platinum foil were used as the reference and counter electrode, respectively. All potentials in this study were measured against the SCE and converted to the RHE reference scale by $E$(V vs. RHE) $= E$(V vs. SCE) $+ 0.0591 \times$ pH $+ 0.241$. The working electrode was prepared as follows: 10 mg of catalyst powder, 2 ml of isopropyl alcohol, and 80 μl Nafion solution (Sigma Aldrich, 5 wt%) were mixed and sonicated for at least 2 h to form a homogeneous ink. Then, a certain volume of catalyst ink was drop-casted onto glassy carbon electrode with a loading of 0.4 mg cm$^{-2}$. The area of glassy carbon electrode is 1 × 2 cm$^2$ and the practically immersing area in the electrolyte was 1 × 1 cm$^2$. For electrocatalytic $NO_3^-$ reduction, a solution with 0.1 M $K_2SO_4$ and 0.5 M $KNO_3$ was used as the electrolyte unless otherwise specified and was evenly distributed to the cathode and anode compartment. The electrolyte volume in the two parts of H-cell was 25 mL and was purged with high-purity Ar for 10 min before the measurement. The LSV was performed at a rate of 5 mV s$^{-1}$. The potentiostatic tests was conducted at constant potentials for 0.5 h at a stirring rate of 500 r.p.m. High-purity Ar was continuously fed into the cathodic compartment during the experiments. Solution resistance ($R_s$) was determined by potentiostatic electrochemical impedance spectroscopy (PEIS) at frequencies ranging from 0.1 to 200 kHz. For consecutive recycling test, the potentiostatic tests were performed at −0.66 V for 0.5 h at a stirring rate of 500 r.p.m. After electrolysis, the electrolyte was analyzed by UV–Vis spectrophotometry as mentioned below. Then, the potentiostatic tests were carried out at the same conditions using the fresh electrolyte for the next cycle. For electrochemical flow cell tests, typically 0.4 mg cm$^{-2}$ Fe SAC and 0.5 mg cm$^{-2}$ $IrO_2$ were air-brushed onto two Sigracet 39 BC GDL (Fuel Cell Store) electrodes as nitrate reduction cathode and oxygen evolution reaction anode, respectively. The two electrodes were placed on opposite sides of two 0.5-cm-thick PTFE sheets with 0.5 cm wide by 2.0 cm long channels so that the catalyst layer interfaced with the flowing liquid electrolyte. A bipolar membrane (Fuel Cell Store) was used to separate the anode and cathode. The anode is circulated with 1.0 M KOH electrolyte at 3 mL min$^{-1}$ flow rate while the flow rate of the 0.5 M $KNO_3$/0.1 M $K_2SO_4$ in the middle flow channel is 1 mL min$^{-1}$. A saturated calomel electrode was connected to cathode channel as the reference electrode. All of the measured potentials were manually 50% compensated. All of the current densities reported in this work are based on geometric surface area.

**Calculation of the FE and NH$_3$ yield rate**. The FE of electrocatalytic $NO_3^-$–$NH_3$ conversion and $NO_3^-$–$NO_2^-$ conversion was calculated as follows:

$$FE_{NH_3} = (8 \times F \times C_{NH_3} \times V)/(17 \times Q) \quad (2)$$

$$FE_{NO_2^-} = \left(2 \times F \times C_{NO_2^-} \times V\right)/(46 \times Q) \quad (3)$$

The rate of NH$_3$ yield rate was calculated using the following equation:

$$r_{NH_3} = (C_{NH_3} \times V)/(t \times m_{cat.}) \quad (4)$$

where $F$ is the Faraday constant (96,485 C mol$^{-1}$), $C_{NH_3}$ is the measured NH$_3$ concentration, $V$ is the volume of the cathodic electrolyte, $Q$ is the total charge

passing the electrode, $t$ is the reduction time, and $m_{cat.}$ is the loading mass of catalysts.

**Determination of ammonia**. The concentration of produced NH$_3$ was spectrophotometrically determined by the indophenol blue method with modification[66]. First, a certain amount of electrolyte was taken out from the electrolytic cell and diluted to the detection range. Then, 2 mL of solution was removed from the diluted electrolyte. Subsequently, 2 mL of a 1 M NaOH solution containing 5 wt% salicylic acid and 5 wt% sodium citrate was added to the aforementioned solution, followed by the addition of 1 mL of 0.05 M NaClO and 0.2 mL of 1.0 wt% $C_5FeN_6Na_2O$ (sodium nitroferricyanide) solution. After 2 h at room temperature, the absorption spectrum was measured by using a UV–vis spectrophotometer (UV-2600). The formation of indophenol blue was determined using the absorbance at a wavelength of 655 nm. The concentration–absorbance curve was made using a series of standard ammonium chloride solutions.

**Determination of nitrite[22]**. Firstly, 0.2 g of N-(1-naphthyl) ethylenediamine dihydrochloride, 4 g of p-aminobenzenesulfonamide, and 10 mL of phosphoric acid ($\rho = 1.685$ g mL$^{-1}$) were added into 50 mL of deionized water and mixed thoroughly as the color reagent. When testing the electrolyte from electrolytic cell, it should be diluted to the detection range. Then 5 mL of the diluted electrolyte and 0.1 mL of color reagent were mixed together. After 20 min at room temperature, the absorption spectrum was measured by using a UV–vis spectrophotometer (UV-2600), and the absorption intensity was recorded at a wavelength of 540 nm. A series of standard potassium nitrite solutions were used to obtain the concentration–absorbance curve by the same processes.

**NMR determination of ammonia**. The NH$_3$ concentration was also quantitatively determined by $^1H$ nuclear magnetic resonance (NMR, 500 MHz) with using DMSO-$d_6$ as a solvent and maleic acid ($C_4H_4O_4$) as the internal standard. The calibration curve was made as follows. First, a series of ammonium chloride solutions with known concentration were prepared in 0.01 M HCl containing 0.5 M $KNO_3$ as standards; second, 0.5 mL of the standard solution was mixed with 0.1 mL DMSO-$d_6$ (with 0.04 wt% $C_4H_4O_4$; 20 mg $C_4H_4O_4$ in 50 g DMSO-$d_6$); third, the mixture was tested by a 500 MHz SB Liquid Bruker Avance NMR spectrometer at room temperature; finally, the calibration curve was achieved using the peak area ratio between NH$_4^+$ and $C_4H_4O_4$ because the NH$_4^+$ concentration and the area ratio are positively correlated. For testing the produced NH$_4^+$ from $NO_3^-$ reduction, the pH of obtained electrolyte must be adjusted to 2.0 before the test. Then, the processes of testing produced NH$_4^+$ are the same to that for making the calibration curve. The amount of produced NH$_4^+$ can be calculated from the peak area using the calibration curve.

**$^{15}$N isotope-labeling experiment**. An isotope-labeling experiment using 0.10 M $K_2SO_4$/0.50 M K$^{15}NO_3$ (98 atom% $^{15}$N) mixed solution as the electrolyte was carried out to clarify the source of NH$_3$. After $^{15}NO_3^-$ electroreduction for 0.5 to 2 h at −0.66 V (vs. RHE), the obtained $^{15}NH_4^+$ was tested by $^1H$ nuclear magnetic resonance (NMR, 500 MHz). The NMR test method of $^{15}NH_4^+$ is the same to that of $^{14}NH_4^+$.

**Computational details**. Atomic simulation environment (ASE) was used to handle the simulation[67]. All electronic structure relaxations were performed using QUANTUM ESPRESSO code[68]. The electronic wavefunctions were expanded in plane waves with a cutoff energy of 500 eV while 5000 grids were used for electronic density representation. To approximate the core electrons ultrasoft pseudopotentials were adapted[69]. Perdew–Burke–Ernzerhof (PBE) exchange-correlation functional was used to calculate the adsorption energies[70]. A one-layer two-dimensional graphene structure was used with a 5 × 5 super cell lateral size. The periodic images were separated by adding a vacuum of 18 Å. Additional layers of graphene have been shown to have negligible effect on the adsorption energies of the intermediates[71]. A (4 × 4 × 1) Monkhorst–Pack k-point was used to sample the Brillouin zone. We apply computational hydrogen electrode method to calculate the adsorption free energies, which assumes the chemical potential of an electron–proton pair is equal to that of ½ $H_2$ in the gas phase. The free energies of adsorption are then calculated as $\triangle G = \triangle E_{DFT} + \triangle$(ZPE − TS), where $\triangle E_{DFT}$, ZPE, T, and S are adsorption enthalpy, zero-point energy, temperature, and entropy, respectively. The limiting potential is calculated by taking the negative of the maximum free energy difference between each two successive steps in the free energy diagram.

Adsorption free energies are calculated using HNO$_3$ as a reference suggested by Calle-Vallejo et al.[72]. We applied 1.12 eV correction to compensate for the DFT error of calculated formation energy of HNO$_3$ (ref. [72]). We investigated the effect of solvation on the adsorption energies of the critical step NO* reduction to HNO* using an optimized explicit solvation model (Supplementary Fig. 53). This analysis showed a negligible change in the calculated limiting potential due to the solvent interaction.

## Data availability

The data that support the plots within this paper and other findings of this study are available from the corresponding authors upon reasonable request.

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

## Acknowledgements

This work was supported by Rice University, the National Science Foundation Nanosystems Engineering Research Center for Nanotechnology Enabled Water Treatment (NEWT EEC 1449500), and the Welch Foundation Research Grant (C-2051-20200401). H.W. is a CIFAR Azrieli Global Scholar in the Bio-inspired Solar Energy Program. S.S. acknowledges the support from the University of Calgary's Canada First Research Excellence Fund Program, the Global Research Initiative in Sustainable Low Carbon Unconventional Resources. Aberration-corrected STEM-EELS was conducted at the Center for Nanophase Materials Sciences, which is a DOE Office of Science User Facility. The authors acknowledge Prof. H.W. Liang and M.X. Chen for XAS measurement and data analysis.

## Author contributions

Z.-Y.W. and H.W. conceptualized the project. H.W. and S.S. supervised the project. Z.-Y.W. developed and performed catalyst synthesis. Z.-Y.W., Q.H., F.-Y.C., and C.X. conducted the catalytic tests of catalysts and the related data processing. Z.-Y.W. performed materials characterization with the help of D.A.C., Q.X., M.S., J.Y.K., Y.X., K.H., and Y.H. [1]H NMR experiments and analysis was carried out by P.Z., Z.-Y.W., and Q.Z.H. S.S., M.K., X.Y., and I.G. performed the DFT simulation. Z.-Y.W., H.W., and S.S. wrote the manuscript. M.S.W. and Q.L. helped the revision of the manuscript. All authors discussed the results and commented on the manuscript.

## Competing interests

The authors declare no competing interests.
