## [Peer Review File · Nature Communications]

Reviewers' comments:

Reviewer #1 (Remarks to the Author):

In this work, the authors reported electrochemical reduction of nitrate into ammonia on Fe SAC. The maximal Faradaic efficiency was up to 75% and ammonia production rate was up to $\sim 20,000 \text{ ug h}^{-1} \text{ mg cat.}^{-1}$. They claimed that the isolate nature of Fe could effectively prevent the N-N coupling, thus switching off the side reactions. The paper is carefully prepared and well written. However, some descriptions and result explanations are not satisfactory. I would suggest that they should address the following major concerns before reconsidering its acceptance.

1. Electrochemical reduction to remove nitrate contaminants has potential application prospects in wastewater treatment. However, ammonia is a suitable reducing product as NH_3 itself is also believed to be a water pollutant and the reduction of nitrate to ammonia requires more energy than that to nitrogen.

2. On page 12, the authors stated that “the main byproduct of nitrate reduction on Fe SAC is NO_2^- suggesting that NO_2^- could be an intermediate product and can be further reduced to NH_3 under more negative potentials”. However, in Figure 4, reduction of HNO_2 to NO was predicted to be thermodynamically favorable, indicating that NO_2^- would be unstable.

3. P16, their DFT calculation showed that the reduction steps of NO^* to HNO^* and HNO^* to N^* would be the rate limiting steps, indicating that NO and HNO would be long-life intermediates. However, lack of experimental evidences supported such species.

4. “the rate limiting steps actually are the NO^* protonation step to HNO^* and HNO^* protonation step to N^* based on the DFT calculations.” These two steps are H^+/e^- reduction rather than protonation processes.

5. On page 17, “We note here that due to the single atom nature of active sites in our catalyst, it is energetically unfavorable to make N-N coupling intermediates or products such as N_2O ...”. Why only considered N_2O ? How about other N-N coupling products, such as $\text{NH}_3\text{-NO}_2^-$, ON-NO_2^- ?

6. On page 17, “As it can be seen the potential determining step is the protonation of NO^* to HNO^* on both Co and Ni SAC. This step on Co and Ni SAC has higher energy barriers (0.42 eV and 0.39 eV, respectively) than the one on Fe SAC (0.30 eV) ...” It is inappropriate to use “energy barrier” to describe the free energy difference between different intermediates.

Reviewer #2 (Remarks to the Author):

The authors use a single atom Fe catalyst for electrocatalytic NO_3^- -to- NH_3 conversion. Nitrate reduction reaction (NO_3R) has typically been done on metals and alloys, but this is the first study I know of that uses single atom catalysts. Additionally, Fe is earth abundant, so this strategy has merit to make a potential impact for practical nitrate treatment in wastewater.

The NH₃ Faradaic efficiency (FE) of ~75% at -0.66 V versus reversible hydrogen electrode (vs. RHE) is quite good. It is debatable if the current densities are good because they are not normalized to electrochemically active surface area as far as I can tell. Still, the analysis and experiments in this work are quite sound and give a number of useful insights for the field. I believe this work could potentially be publishable in Nature Comm. I have a few questions and comments that should first be considered, which I hope the author's find useful to improve their work.

- Pg 11: Did the authors use geometric surface area when reporting current densities? This aspect can be clarified.
- Pg 11: 0.4 mg cm⁻² seems to be pretty high loading; are the authors sure they are depositing a thin enough layer to prevent internal mass transport limitations?
- Pg 12: Potential concern: I thought that 14NH₄⁺ peaks typically come from contaminants and potential N in the catalyst. If the 14NH₄⁺ peaks are lining up with UV-Vis, it's telling me that the ammonia formed can also be nitrogen contributed from the catalyst (since I believe the catalyst as a single Fe surrounded by N atoms). No NH₃ is said to be detected if KNO₃ is not around, but the authors should provide the proof somewhere (i.e., by adding such data to the supporting information).
- Pg 13: At pH 13, Fe might start to turn into Fe(OH)₂ at the proposed operating potential? Or is this not a concern for the single atom Fe¹?
- Pg 14: How did the authors normalize by metal content? Just by the wt% of the material? Dispersion and ECSA would provide more accurate normalizations and comparisons.
- SI: Did the authors ever measure the concentration of NO₃ with UV-Vis? I was under the impression that NO₂ spectra overlaps with NO₃ so the two need to be deconvoluted.
- Pg 16: "While NO₂⁻ is confirmed as an intermediate product in the experimental result, the rate limiting steps actually are the NO* protonation step to HNO* and HNO* protonation step to N* based on the DFT calculations."

- How does this finding compare with other calculations and measurements? Typically nitrate adsorption/dissociation is the rate-determining step on metals. Perhaps this aspect can be mentioned, since it is different than what is typically proposed for nitrate reduction.

I also recommend the authors be careful when using the language rate determining step vs potential determining step when discussing their modeling results. Activation barriers are not computed nor a microkinetic model constructed, just reaction thermodynamics reported at some applied potential. Thus, I feel only "potential limiting step" should be used in this discussion and conclusions.

e.g. in conclusions: "Our DFT simulations reveal the reaction pathways and rate limiting steps of nitrate reduction on Fe single atomic site." I think this would potential limiting steps based on the data and analysis provided.

More computational details in the Methods should be provided:

(1) Are solvent effects considered implicitly or not treated? Or is only nitrate(aq) treated but all other species are in vacuum?

(2) How was the applied potential treated? Using the computational hydrogen electrode? This approach was not described in the Computational Methods or paper as far as I could see.

(3) Was the 1-layer graphene structured allowed to relax upon adsorption of species or were its xyz coordinates fixed? What evidence is there that 1 layer graphene is sufficiently accurate of a model and does not suffer finite-size effects? What would happen if the graphene was 2 or 3 layers thick?

Figure 4 Caption: “(b) Free energy diagram showing the minimum energy pathway at $U = 0.0$ V (green) and calculated limiting potential of -0.30 V (black).”

- Would clarify to specify the reference of the voltage in figure caption, e.g., vs. RHE.

Reviewer #3 (Remarks to the Author):

The manuscript claims that Iron-single atoms can effectively reduce nitrate to ammonia with high selectivity and efficiency along with a high yield rate. It can not be accepted in the current form due to the following:

1. A claim of high selectivity should be more clearly demonstrated apart from Supplementary Fig. 35, which does not compare Fe-SAC

2. The claim about high efficiency is not new: No conversion efficiency of nitrate to nitrite/ammonia/nitrogen/H₂ was reported; Lack of knowledge about the field, just because Fe-SACs haven't been studied before doesn't make it a novel material unless the authors can substantiate the claim about performance with concrete corroborating evidence; No comparison with the past literature was created to demonstrate the novelty or better performance of their work. A brief comparison is provided below, with just one literature report cited within the manuscript, and their results don't even come at par with this study except for the high yield

Performance Metrics This Work Literature (10.1002/ange.201915992)

Potential vs -0.85 V -0.85

Faradaic Efficiency $\sim 65-70\%$ 95.8%

Conversion Not reported 97%

Selectivity Not reported 81.2%

The above performance metrics are well established in literature to demonstrate the performance of any electrode material. Clearly, the researchers haven't done a thorough literature review. Nitrate reduction to ammonia is more actively researched by those working in water treatment, and most of the research is better accessible with keywords like “water treatment,” “wastewater treatment”, “nitrate removal”, “denitrification”, etc.

3. It should be noted that iron has already been reported as highly selective for nitrate reduction to ammonia in 2005/2009. The efficiency is also similar to the one reported in this study.

- 10.1016/j.watres.2005.07.032

- 10.1016/j.electacta.2009.03.064

The focus of wastewater treatment is not ammonia generation, but in-fact nitrogen removal; therefore, the ammonia yield rate is not actively reported in these studies about nitrate reduction.

If the authors want to demonstrate that Fe-SAC is better than iron-based bulk metal electrodes, they must, at the least, demonstrate comparison with more than just iron nanoparticles. For example, iron mesh, foam, foil, etc.

4. Moreover, in supplementary Fig. 35, they show FE of nitrate reduction to NH_3 and NO_2^- for NC and FeNP/NC. If you superimpose the subgraphs (a) on to (b), we observe barely any difference in nitrite reduction efficiency. Whereas for FeNP/NC has a curve for FE- NH_3 slightly translated in the y-axis but follows the same trend. Which begs the question: How well-controlled were these two experiments?

5. Overall this article does not bring any significant contribution to the field. Established literature has already reported better efficiencies and an understanding of the mechanism for nitrate reduction. An insight into the high yield would perhaps be of significant interest, which has not been critically analyzed.

6. More control experiments need to be conducted to address the following:

- NC catalyst showed ammonia production Fig 3c

o A control experiment to ensure that the ammonia produced is indeed nitrate reduction rather than trapped ammonia from the synthesis process (ammonium peroxydisulphate was used in synthesis).

7. There are so many different approaches to synthesize Fe-SACs, especially those doped on a carbon support. Since SiO_2 was used as a template, were any efforts made to confirm the complete removal of Si from the prepared catalyst. Was an XPS scan for Si made during the characterization process to ensure no trace amounts of Si atoms remained on the catalyst?

8. A control experiment with a Fe-SAC on carbon support through another synthesis route without any silicon presence should be carried out to ensure there was no interference from any trace amounts of the Silicon template. (A good practice when working with single-atoms is to minimize the no. of elements used in the synthesis process to ensure minimal contamination)

9. DFT Simulations.

Some packages allow for simulating the solvation effect and pH effect on DFT calculations; they should be considered to provide a deeper understanding of the reaction mechanism.

Response to reviewers' comments:

Reviewer 1

In this work, the authors reported electrochemical reduction of nitrate into ammonia on Fe SAC. The maximal Faradaic efficiency was up to 75% and ammonia production rate was up to ~ 20,000 ug h⁻¹ mg_{cat}⁻¹. They claimed that the isolate nature of Fe could effectively prevent the N-N coupling, thus switching off the side reactions. The paper is carefully prepared and well written. However, some descriptions and result explanations are not satisfactory. I would suggest that they should address the following major concerns before reconsidering its acceptance.

--- We appreciate the reviewer's praise of our work, as well as important suggestions which have substantially improved the quality of our manuscript.

1. Electrochemical reduction to remove nitrate contaminants has potential application prospects in wastewater treatment. However, ammonia is a suitable reducing product as NH₃ itself is also believed to be a water pollutant and the reduction of nitrate to ammonia requires more energy than that to nitrogen.

--- We fully understand the reviewer's points here. Just as the reviewer mentioned, electrochemical nitrate reduction with N₂ as the targeted product is considered to be a promising method in wastewater treatment field because of its low cost, environmental friendliness, and easy operation. The conversion of nitrate to N₂ is really important and meaningful in a viewpoint for the removal of nitrate pollutants. However, N₂ is a valueless product compared with high value-added NH₃. **The aim of this work is to convert the nitrate pollutants into valuable NH₃, not complete removal of pollutants.** Additionally, although NH₃ can be a water pollutant, it could be easily separated by extraction technology and regenerated resins when the concentration is high. In addition, low-concentration ammonia could also be effectively separated by air stripping method after pH adjustment [Waste Manage. 2003, 23, 441-446]. But separation of nitrate contaminants are highly difficult. On the other hand, NH₃ produced by nitrogen reduction is also in a form of aqueous solution, which is considered as a valuable product, not a pollutant.

We agree with the reviewer's viewpoint that the reduction of nitrate to ammonia requires more energy than that to nitrogen. What we would like to express is that the energy consumed for converting nitrate to NH₃ should be cost-efficient, considering industrial-scale NH₃ synthesis from N₂ and H₂, i.e. Haber-Bosch process, requires harsh operating conditions including high temperature (400-500 °C) and high pressure (150-300 atm), and causes serious CO₂ emissions.

More importantly, the electricity used in this process can be generated from clean/renewable energies now, such as solar and wind. Thus, electrochemically converting nitrate back to valuable ammonia can be a green and promising route for ammonia synthesis.

2. On page 12, the authors stated that “the main byproduct of nitrate reduction on Fe SAC is NO_2^- suggesting that NO_2^- could be an intermediate product and can be further reduced to NH_3 under more negative potentials”. However, in Figure 4, reduction of HNO_2 to NO was predicted to be thermodynamically favorable, indicating that NO_2^- would be unstable.

--- Thanks for this important comment here. The point that NO_2^- could be an intermediate product and can be further reduced to NH_3 under more negative potentials was validated by the experiment of NO_2^- reduction on Fe SAC. We found that more than 90% FE of NH_3 and higher production rates could be achieved under the studied potential window for NO_2^- reduction on Fe SAC (Supplementary Fig. 29), which indicated that NO_2^- reduction was much easier than NO_3^- reduction. The NO_2^- as an intermediate product of NO_3^- reduction have also been observed in many other studies [Catal. Commun. 2009, 10, 1975-1979; J. Electroanal. Chem. 2009, 630, 69-74; Natl. Sci. Rev. 2019, 6, 730-738].

We also would like to kindly draw the reviewer attention to the following two electrochemical steps for Figure 4: $^*\text{NO}_2 + (\text{H}^+ + \text{e}^-) \rightarrow ^*\text{HNO}_2$; $^*\text{HNO}_2 + (\text{H}^+ + \text{e}^-) \rightarrow \text{NO}^* + \text{H}_2\text{O}$. We agree with the reviewer that reduction of $^*\text{HNO}_2$ to $^*\text{NO}$ is thermodynamically favorable. However, formation of $^*\text{HNO}_2$ from $^*\text{NO}_2$ is exergonic indicating that reduction of $^*\text{NO}_2$ to ammonia through subsequent reaction steps does not occur before applying negative potentials that make all subsequent steps downhill in free energy. During nitrate reduction process, parts of NO_2^* could desorb from the surface of Fe SAC and form the byproduct of NO_2^- . Thus, NO_2^- could be an intermediate product in the nitrate reduction on Fe SAC.

3. P16, their DFT calculation showed that the reduction steps of NO^* to HNO^* and HNO^* to N^* would be the rate limiting steps, indicating that NO and HNO would be long-life intermediates. However, lack of experimental evidences supported such species.

--- We thank the reviewer for the constructive comment here. In catalysis community, it is very important to employ experimental methods to investigate catalysis processes and understand reaction mechanisms. Operando and in-situ studies are mainly used methods, which involve advanced instruments and complicated devices as well as sophisticated experiment operation. Up to now, it is still highly challenging to directly observe specific reaction intermediates and study mechanisms by operando and in-situ studies, because of complicated multi-phase interfaces during electrochemical processes, and insufficient sensitivity of instruments. Since DFT calculation can effectively simulate the electrochemical reactions, now DFT calculations are widely used to study reaction mechanisms, and to identify reaction intermediates and potential limiting steps. Single atom catalysts (SACs) have well-defined atomic structures, which can serve as a great platform for studying of reaction pathways. Thus, in this study, we used DFT calculation to investigate the reaction mechanisms

for nitrate reduction on Fe SAC. Our DFT calculation revealed the potential limiting steps are reduction of NO* to HNO* and HNO* to N*. Since reduction of NO* to HNO* as well as the reduction of HNO* to N* are electrochemical steps, the energy barrier associated with these steps can be overcome when a potential that is equal to the free energy difference of the most exergonic step is applied that is:

Where ΔG_1 , ΔG_2 are the changes in free energies of the respective electrochemical steps, *i.e.*, steps 1, 2, .. *etc.* at $U = 0$ vs Reversible Hydrogen Electrode (RHE). We agree with the reviewer that our current experiments do not consider identifying intermediates involved in nitrate reduction reaction, because now we do not have feasible methods or experimental instruments to investigate the intermediates of the nitrate reduction on Fe SAC. However, almost all previous studies on transition metals agree that NO* is a key intermediate and its reduction to HNO* or NOH* is a key for driving the reaction to ammonia product [ACS Catal. 2017, 7, 4660-4667; Angew. Chem. Int. Ed. 2015, 54, 8255-8258; J. Am. Chem. Soc. 2007, 129, 10171-10180; J. Electroanal. Chem. 2003, 554-555, 15-23; ACS Catal. 2018, 8, 503-515; Small Methods 2020, 4, 2000672; ACS Catal. 2019, 9, 7957-7966].

Additionally, we have other two points that we would like to highlight. First, the intermediate before the potential limiting steps might not necessary to be observed/detected during the reactions. One example to demonstrate this point is the ORR on Pt surface, where the last reduction step, *OH to H₂O, was identified as the potential limiting steps [Science 2016, 354, 1031], but people have never observed the OH radical as the intermediate products during ORR. Secondly, the observed intermediates cannot guarantee they are from the potential limiting steps. An example that we want to mention here is the CO₂ reduction to C₂ products on Cu, where CO can be detected as the reaction intermediate. However, instead of the next OC-CO coupling step as the rate limiting step, in previous studies people discovered that the rate limiting step was the OCCO protonation to OCCHO step [Nat. Catal. 2018, 1, 111]. Anyway, we believe the reviewer provided us a highly important and constructive comment here, which helped us rethink how to understand the reaction mechanism. We will bear this valuable comment in our mind for our further catalysis research. Thanks again!

4. *“the rate limiting steps actually are the NO* protonation step to HNO* and HNO* protonation step to N* based on the DFT calculations.” These two steps are H⁺/e⁻ reduction rather than protonation processes.*

--- Many thanks for the reviewer’s very important comment here. We corrected this sentence as well as everywhere else in the manuscript to reflect the reviewer’s point. We used the term reduction instead of protonation to reflect the fact that the step involves transferring a coupled H⁺/e⁻. The following sentence is included in the revised version of our manuscript. *“our DFT calculations show that the potential limiting steps are the NO* reduction to HNO* and HNO* reduction to N*...”*

5. On page 17, “We note here that due to the single atom nature of active sites in our catalyst, it is energetically unfavorable to make N-N coupling intermediates or products such as N_2O ...”. Why only considered N_2O ? How about other N-N coupling products, such as $NH_3-NO_2^-$, $ON-NO_2$?

--- We appreciate the reviewer’s important comment here. We investigated the N_2O because it is the likely intermediate in the reaction mechanism for the N_2 production, a competing pathway for ammonia synthesis. Additionally, apart from N_2 , N_2O is the smallest possible product/intermediate that may form due to N-N coupling. We expect the coupling of larger intermediates to be highly unfavorable at the single site due to the spherical hindrance and the small space available for binding at the single site. The thermodynamic free energies of species such as $NH_3-NO_2^-$ and $ON-NO_2$ are not known which makes it difficult to make a comprehensive conclusion. Also, these species have not been suggested as likely intermediates in N_2 formation reaction mechanism.

6. On page 17, “As it can be seen the potential determining step is the protonation of NO^* to HNO^* on both Co and Ni SAC. This step on Co and Ni SAC has higher energy barriers (0.42 eV and 0.39 eV, respectively) than the one on Fe SAC (0.30 eV) ...” It is inappropriate to use “energy barrier” to describe the free energy difference between different intermediates.

--- We appreciate the reviewer for this suggestion and modified the sentence in the revised version of our manuscript: “As it can be seen the potential limiting step is the reduction of NO^* to HNO^* on both Co and Ni SACs. The calculated limiting potentials for nitrate reduction on Co and Ni, and Fe SACs are 0.42, 0.39, and 0.3 V, respectively...” .

Reviewer 2

The authors use a single atom Fe catalyst for electrocatalytic NO_3^- -to- NH_3 conversion. Nitrate reduction reaction (NO_3R) has typically been done on metals and alloys, but this is the first study I know of that uses single atom catalysts. Additionally, Fe is earth abundant, so this strategy has merit to make a potential impact for practical nitrate treatment in wastewater. The NH_3 Faradaic efficiency (FE) of ~75% at -0.66 V versus reversible hydrogen electrode (vs. RHE) is quite good. It is debatable if the current densities are good because they are not normalized to electrochemically active surface area as far as I can tell. Still, the analysis and experiments in this work are quite sound and give a number of useful insights for the field. I believe this work could potentially be publishable in Nature Comm. I have a few questions and comments that should first be considered, which I hope the author’s find useful to improve their work.

--- We highly appreciate the reviewer's positive comments on our work, as well as the important suggestions which have substantially improved the quality of our manuscript.

1. Pg 11: Did the authors use geometric surface area when reporting current densities? This aspect can be clarified.

--- Yes all of the current densities reported in this work are based on geometric surface area. We have clarified this point in the methods part of this manuscript (Page 21).

2. Pg 11: 0.4 mg cm⁻² seems to be pretty high loading; are the authors sure they are depositing a thin enough layer to prevent internal mass transport limitations?

--- We thank the reviewer for this point. The catalyst loading amount of 0.4 mg cm⁻² is actually a common loading typically used in carbon-based materials for electrochemical tests. In general, the carbon-based catalyst loading amount is within 0.2 to 1 mg cm⁻², for example, 0.6 mg cm⁻² for atomically dispersed Fe³⁺-N-C catalyst [Science 2019, 364, 1091-1094], 0.8 mg cm⁻² for atomically dispersed Mn-N-C catalyst [Nat. Catal. 2018, 1, 935-945], 0.6 mg cm⁻² for (CM+PANI)-Fe-C catalyst [Science 2017, 357, 479-484], and 0.275 mg cm⁻² for MN₄C₄ (M = Fe, Co, Ni) single-atom catalysts [Nat. Catal. 2018, 1, 63-72]. We believe that the deposited Fe SAC on glassy carbon is thin enough layer, which can prevent internal mass transport limitations.

3. Pg 12: Potential concern: I thought that ¹⁴NH₄⁺ peaks typically come from contaminants and potential N in the catalyst. If the ¹⁴NH₄⁺ peaks are lining up with UV-Vis, it's telling me that the ammonia formed can also be nitrogen contributed from the catalyst (since I believe the catalyst as a single Fe surrounded by N atoms). No NH₃ is said to be detected if KNO₃ is not around, but the authors should provide the proof somewhere (I.e., by adding such data to the supporting information).

--- Many thanks for reviewer's very important comment and suggestion here. We guess that there might be some confusions or misunderstandings in the ¹H NMR spectra of ¹⁴NH₄⁺ (Figure 3d). As ¹H NMR is not as sensitive as UV-vis for detecting NH₃, the contaminants and N in the catalyst, if there are any that were converted to NH₄⁺, cannot get such strong ¹⁴NH₄⁺ peak on the ¹H NMR spectra. More importantly, if ¹⁴NH₄⁺ peaks observed ¹H NMR spectra came from contaminants and potential N in the catalyst, we should also observe similar ¹⁴NH₄⁺ peaks when we performed nitrate reduction using ¹⁵NO₃⁻. However, only two peaks of ¹⁵NH₄⁺ were observed in ¹H NMR spectra and no any ¹⁴NH₄⁺ peaks were found. In addition, we followed the reviewer's suggestion by providing results of electrocatalysis test without KNO₃. It is found the UV-Vis spectra of electrolyte before and after electrocatalysis are the same and does not show any ammonia generation (Figure R1). All of these results strongly support that the produced NH₃ is from electrochemical nitrate reduction rather than contaminations and

potential N in the catalyst. The results of electrocatalysis test without KNO_3 have been included into our revised manuscript (Supplementary Figure 26).

Figure R1. UV-vis curves of electrolytes before and after electrocatalysis test for Fe SAC in a K_2SO_4 solution without KNO_3 .

4. Pg 13: At pH 13, Fe might start to turn into $\text{Fe}(\text{OH})_2$ at the proposed operating potential? Or is this not a concern for the single atom Fe_1 ?

--- We thank for the reviewer's important comments here. The Fe nanoparticles and bulk Fe could be turned into $\text{Fe}(\text{OH})_2$ during electrocatalysis process at pH 13, but it should not be a concern for the Fe SAC with a Fe-N_4 structure, as the M-N_4 (Fe, Co, Ni) specie in SAC is a very robust coordination structure can tolerate diverse harsh electrocatalysis process for long-term operations, such as acidic ORR [Science 2017, 357, 479-484], alkaline ORR [Proc. Natl. Acad. Sci. USA 2018, 115, 6626-6631], alkaline OER [Nat. Catal. 2018, 1, 63-72], CO_2RR at very negative potential (-0.5 V vs RHE) in KHCO_3 [Science 2019, 364, 1091-1094], and alkaline HER [Nat. Catal. 2019, 2, 134-141; J. Phys. Chem. Lett. 2020, 11, 6691-6696] as well as nitrate reduction in neutral pH at very negative potential (-0.66 V vs RHE) in this work. To clearly demonstrate this point, we chose a recent published work about using the Fe_1/NC for alkaline HER as an example [J. Phys. Chem. Lett. 2020, 11, 6691-6696], which is similar to our test conditions at pH 13. In the study, the authors performed operando X-ray absorption spectroscopy to reveal the structure change of Fe-N_4 active sites during the catalysis process at -274 mV in 1 M KOH solution. They found that the nature of single atom Fe_1 was well kept with slight change of Fe oxidation state and Fe-N coordination number during the HER catalysis process under these conditions. In addition, the Fe_1/NC could work very well at ca. -110 mV for 20 h, indicating the Fe_1 active sites kept well. Thus, we think turning into $\text{Fe}(\text{OH})_2$ at pH 13 under the proposed operating potential (-0.09V to -0.46 V) should not be a concern for the Fe SAC.

5. Pg 14: How did the authors normalize by metal content? Just by the wt% of the material? Dispersion and ECSA would provide more accurate normalizations and comparisons.

--- We appreciate the reviewer's constructive comments here. Yes we compared NH_3 yield rate of Fe SAC and FeNP/NC normalized by metal contents in our catalysts in Supplementary Figure 38 (revised version). What we want to emphasize in this comparison is to simply demonstrate that Fe single atoms are much more active than that of Fe NPs. Importantly, to avoid any misinterpretation, we mainly compared the ammonia yield rate between Fe SAC and FeNP/NC catalysts based on the overall catalyst loading (0.4 mg/cm^2) in Figure 3c of our main text, which demonstrates the Fe SAC are much more active than FeNP/NC. According to the reviewer's suggestion, we also tested double-layer capacitance (C_{dl}) which is proportional to the ECSA by CV tests. Figure R2 shows that Fe SAC and FeNP/NC have very close C_{dl} , which further demonstrate the intrinsically higher activity of Fe SAC than FeNP/NC. This result have been included into our revised manuscript (Supplementary Figure 39).

Figure R2. Cyclic voltammograms (CV) for (a) Fe SAC and (b) FeNP/NC catalysts at different scan rates from 5 to 40 mV s^{-1} , respectively. Plots showing the extraction of the C_{dl} for (c) Fe SAC and (d) FeNP/NC, respectively.

6. SI: Did the authors ever measure the concentration of NO_3 with UV-Vis? I was under the impression that NO_2 spectra overlaps with NO_3 so the two need to be deconvoluted.

--- Thanks for this important comment. In our study, the concentration of NO_3^- was not tested by UV-vis, but we used UV-vis method to test the concentration of NO_2^- . This detailed method for testing NO_2^- is presented in methods (Page 22). Of note, the method is exclusively effective for testing NO_2^- , and NO_3^- would not have any signal by using the method. Figure R3 show that the KNO_3 solution with a concentration of 0 to $10.0 \mu\text{g mL}^{-1}$ do not exhibit any absorbance at wavelength range from 400 to 700 nm when tested by this method. In a contrast, $2.0 \mu\text{g mL}^{-1}$ KNO_2 can produce a very strong absorbance peak at 540 nm, which can be used for quantify the concentration of NO_2^- .

Figure R3. UV-vis curves of KNO_3 solution with a concentration of 0 to $10.0 \mu\text{g mL}^{-1}$ and $2.0 \mu\text{g mL}^{-1}$ KNO_2 tested by nitrite determination method.

7. Pg 16: “While NO_2^- is confirmed as an intermediate product in the experimental result, the rate limiting steps actually are the NO^* protonation step to HNO^* and HNO^* protonation step to N^* based on the DFT calculations.”

- How does this finding compare with other calculations and measurements? Typically nitrate adsorption/dissociation is the rate-determining step on metals. Perhaps this aspect can be mentioned, since it is different than what is typically proposed for nitrate reduction.

I also recommend the authors be careful when using the language rate determining step vs potential determining step when discussing their modeling results. Activation barriers are not computed nor a microkinetic model constructed, just reaction thermodynamics reported at some applied potential. Thus, I feel only “potential limiting step” should be used in this discussion and conclusions.

e.g. in conclusions: “Our DFT simulations reveal the reaction pathways and rate limiting steps of nitrate reduction on Fe single atomic site.” I think this would potential limiting steps based on the data and analysis provided.

--- We highly appreciate the reviewer’s important comments here. On Page 17 of the original version of our manuscript we mentioned that reduction of NO^* to HNO^* is reported in the literature to be the potential limiting step on metals and we cited the relevant references. This step is also potential limiting in our single site catalyst. To address the reviewer's point, in the

revised version of our manuscript we explained more about the similarities and differences between transition metals and our single site catalysts including the point mentioned by the reviewer about nitrate adsorption/dissociation. The following discussion has been added to the revised version of our manuscript on page 17.

“We find that reduction of NO_2^- to NO is downhill in free energy. This finding is in agreement with a previous computational report on Pd surface. Nitrate reduction on polycrystalline and single crystals of transition metals have been studied in the past. Liu et. al. suggested that N^ and O^* binding energies can be used as descriptors for nitrate reduction performance on metals. In addition, it has been shown that the main product of nitrate reduction reaction on all transition metals is nitrogen with low selectivity towards ammonia. The latter is due to the dominance of parasitic hydrogen evolution reaction. Moreover, NO^* has been suggested as a key intermediate for nitrate reduction on metal surfaces such as Pt where its reduction to HNO^* or NOH^* is a critical step for production of NH_4^+ . Our analysis on Fe SAC shows that NO^* is a key intermediate for nitrate reduction reaction which is consistent with previous computational reports on Pt and Pd. We would like to emphasize that while NO_2^- is confirmed as an intermediate product in the experimental result, our DFT calculations show that the potential limiting steps are the NO^* reduction to HNO^* and HNO^* reduction to N^* in agreement with previous computational reports on transition metals such as Pt and Pd.”*

We have also modified the “rate limiting step” term everywhere in our manuscript with “potential limiting step” to reflect the reviewer's concern.

8. More computational details in the Methods should be provided:

(1) Are solvent effects considered implicitly or not treated? Or is only nitrate (aq) treated but all other species are in vacuum?

--- We thank the reviewer for this comment. Our initial analysis was done without taking into account the solvation effect and our results showed that we can capture the trend observed in the experiment. However, to address reviewer's concern, we investigated the solvent effect on the potential limiting step of Fe SAC that is reduction of NO^* to HNO^* ($\text{NO}^* + (\text{H}^+ + \text{e}^-) \rightarrow \text{HNO}^*$) using an explicit solvation model, which is optimized for our graphene model structure and agrees well with the literature [ACS Cent. Sci. 2017, 3, 1286-1293] (Figure R4). We calculated the change in free energy of the HNO^* formation step from adsorbed NO^* with and without inclusion of explicit water layers. The results are summarized in the Table below indicating a negligible shift (0.001 V) in the calculated change in adsorption free energy of this step as well as the limiting potential as a result of inclusion of the solvation. This small effect won't change the conclusion of this study. The experiment trend can be well captured without including a solvation effect. This analysis along with the figure and above discussion were added to the SI (Supplementary Figure 52).

Electrochemical Step	$\Delta G_{\text{no solvent}}$ (eV)	$\Delta G_{\text{with solvent}}$ (eV)	$U_{\text{L, no solvent}}$ (V)	$U_{\text{L, with solvent}}$ (V)
$\text{NO}^* \rightarrow \text{HNO}^*$	0.278	0.277	0.278	0.277

Figure R4. The optimized structures of (a) NO^* and (b) HNO^* intermediates by inclusion of explicit solvation models.

(2) How was the applied potential treated? Using the computational hydrogen electrode? This approach was not described in the Computational Methods or paper as far as I could see.

--- We thank the reviewer for this comment. Indeed we have used the computational hydrogen electrode approach in our study. We updated the computational details and added more explanation in the revised version to reflect the reviewer's point. The additional changes are highlighted in bold fonts below.

“Atomic Simulation Environment (ASE) is used to handle the simulation and the QUANTUM ESPRESSO program package to perform electronic structure calculations using DFT. The electronic wavefunctions are expanded in a series of plane waves with a cutoff energy of 500 eV and a density cutoff of 5000 eV. Core electrons were approximated using ultrasoft pseudopotentials. To describe adsorption energies, Perdew–Burke–Ernzerhof (PBE) exchange–correlation functional was used. Graphene structures are modeled as one layer with a vacuum of 18 Å to decouple the periodic replicas. **Additional layers of graphene have been shown to have negligible effect on the adsorption energies of the intermediates.** A 5×5 super cell lateral size is used as a model structure and the Brillouin zone is sampled with (4×4×1) Monkhorst–Pack k-points. **All atoms were allowed to relax in x, y and z direction with no constraints. All adsorption configurations were considered and only the most stable ones are reported here. The computational hydrogen electrode method introduced by Nørskov *et al.* was used to calculate the free energy levels of all adsorbates. In this model, the free energy change of each electrochemical reaction step that involves an electron–proton transfer is calculated using the reversible hydrogen electrode (RHE), where the chemical potential of an electron–proton pair is equal to that of half of hydrogen in the gas phase under standard conditions. The electrode potential is taken into account by shifting the electron energy by $-eU$ where e and U are the elementary**

charge and the electrode potential, respectively. The limiting potential is defined as the negative of the maximum free energy difference between any two successive electrochemical steps.

Adsorption free energies are calculated using HNO₃ as a reference suggested by Calle-Vallejo *et al.* taking into account zero-point energy and entropy corrections. We applied 1.12 eV correction to compensate for the DFT error of calculated formation energy of HNO₃. The free energies of adsorptions are calculated as $\Delta G = \Delta E_{DFT} + \Delta(ZPE - TS)$, where ΔE_{DFT} is the calculated electronic adsorption energy in zero Kelvin, ZPE is zero-point energy, S is entropy, and T is temperature. **We investigated the effect of solvation on the adsorption energies of the critical step NO* reduction to HNO* using an optimized explicit solvation model (Supplementary Figure 52). This analysis showed a negligible change in the calculated limiting potential due to the solvent interaction.”**

The changes were also included into revised manuscript (Page 23 in our revised manuscript).

(3) Was the 1-layer graphene structured allowed to relax upon adsorption of species or were its xyz coordinates fixed? What evidence is there that 1 layer graphene is sufficiently accurate of a model and does not suffer finite-size effects? What would happen if the graphene was 2 or 3 layers thick?

--- We thank the reviewer for this comment. We have used the two-dimensional graphene model in our calculation and allowed atoms to relax in x, y and z direction with no constraints. We have not taken into account any extra layers of graphene in our calculations because it has been reported in the literature [Nano Res. 2017, 10, 1163-1177] that the additional graphene layer has a negligible effect on the adsorption energies. This is because the graphene layers are ~3.5 Å apart and they hardly have any effect on the adsorption energies of the intermediates. We expect the additional graphene layer would increase the computational cost but would only have a negligible but constant impact on the adsorption energies of all intermediates. Therefore, the conclusions of our study remain unchanged.

9. *Figure 4 Caption: “(b) Free energy diagram showing the minimum energy pathway at U = 0.0 V (green) and calculated limiting potential of -0.30 V (black).”*

- Would clarify to specify the reference of the voltage in figure caption, e.g., vs. RHE.

--- We thank the reviewer for noticing this point and we have now modified the caption.

Reviewer 3

The manuscript claims that Iron-single atoms can effectively reduce nitrate to ammonia with high selectivity and efficiency along with a high yield rate. It cannot be accepted in the current form due to the following:

--- We highly appreciate the reviewer's time and efforts in reviewing our manuscript, and providing us constructive comments and suggestions to further improve the quality of our paper. In this new version of manuscript, we have addressed all questions raised by reviewer regarding both the experimental and theoretical analysis, included additional characterizations and discussions, which has greatly improved the depth and rigor of this work.

1. A claim of high selectivity should be more clearly demonstrated apart from Supplementary Fig. 35, which does not compare Fe-SAC.

--- We thank reviewer's important comment here. The NH₃ selectivity (i.e. NH₃ FE) of Fe SAC is presented in our main text, Figure 3b. We have also clearly discussed the performance result, compared to NC and FeNP/NC in pages 11 and 14 of our main text. To avoid repeatedly report the data, we thus did not present the NH₃ selectivity of Fe SAC in Supplementary Fig. 35 (Supplementary Fig. 37 in revised Supplementary Information). Besides, the Supplementary Note 3 (Page 61 in Supplementary Information) clearly discussed the higher selectivity as well as activity of Fe SAC than those of Co SAC and Ni SAC.

2. The claim about high efficiency is not new: No conversion efficiency of nitrate to nitrite/ammonia/nitrogen/H₂ was reported; Lack of knowledge about the field, just because Fe-SACs haven't been studied before doesn't make it a novel material unless the authors can substantiate the claim about performance with concrete corroborating evidence; No comparison with the past literature was created to demonstrate the novelty or better performance of their work. A brief comparison is provided below, with just one literature report cited within the manuscript, and their results don't even come at par with this study except for the high yield

Performance Metrics This Work Literature (10.1002/ange.201915992)

Potential vs -0.85V -0.85

Faradaic Efficiency ~65-70% 95.8%

Conversion Not reported 97%

Selectivity Not reported 81.2%

The above performance metrics are well established in literature to demonstrate the performance of any electrode material. Clearly, the researchers haven't done a thorough

literature review. Nitrate reduction to ammonia is more actively researched by those working in water treatment, and most of the research is better accessible with keywords like “water treatment,” “wastewater treatment”, “nitrate removal”, “denitrification”, etc.

--- We completely understand the reviewer’s concern here, but we would like to express that we fully understood that most of nitrate reduction works have been in the environmental wastewater treatment field, and we have done very through literature review. We can easily include a performance comparison table as what we listed below, but there are several reasons why we did not include it in our previous manuscript:

1. The target products are different. In traditional wastewater treatment field, most of the works have been focused on reducing nitrate to N_2 , and ammonia is their byproduct. However, in our study, our target is to convert the nitrate waste into valuable ammonia.
2. The electrochemical characterization is different. In traditional nitrate reduction research, most of the studies report “conversion efficiency” and “selectivity”. These two terms are different from “Faradaic efficiency” we typically use in multi pathway electrocatalysis such as CO_2 reduction or N_2 reduction [Water Res. 2010, 44, 1918-1926; Electrochim. Acta 2019, 324, 134846; Nanoscale 2018, 10, 19023-19030; Nature 2020, 581, 178-183; Adv. Mater. 2018, 30, 1803498]. In traditional work, they usually quantify the concentration of nitrate before and after catalysis, as well as the ammonia concentration after catalysis, which can give out the N_2 generation. Other side products such as H_2 are not taken into consideration as this pathway is not within nitrate conversion. However in our study, we use Faradaic efficiency to quantify the ammonia and other side products’ selectivity, which is based on the overall charges passed into the electrochemical system. The “Faradaic efficiency” reported here and the “selectivity” reported before are therefore not be comparable.
3. In our current study, as well as many standard electrocatalysis studies, we do not typically characterize the reactant conversion efficiency. What we usually do is to ensure enough reactants (such as enough concentration or nitrate in this study, or enough flow rate of pure CO_2 stream in CO_2 reduction) to evaluate the intrinsic catalytic activity of our catalyst, which helps to avoid any mass diffusion limits which could impact the apparent performance of the catalyst. Therefore in this work the consumed nitrate during our performance evaluation is negligible, maintaining a stable nitrate concentration. When the catalytic materials are ready for large-scale practical applications, more detailed and technical questions such as the conversion efficiency or one-pass efficiency could be evaluated, which will be a balance between catalyst’s intrinsic performance and mass diffusion limits. While in previous wastewater treatment studies, people usually convert a significant portion of nitrate ions in solution for quantification and high value of conversion. When the remaining nitrate concentration becomes extremely low, typically the hydrogen evolution reaction will take off, which however won’t be counted into “selectivity” as this is not nitrate conversion, but will be counted into “Faradaic efficiency” as it could occupies majority

of the input electricity. The purpose of these studies is different therefore a direct comparison is not appropriate.

Following the reviewer's important suggestion here, we carefully made a table including the performance of Fe SAC as well as previously reported catalysts for electrocatalytic nitrate reduction, which are mainly from water treatment, as kindly suggested by reviewer. From Table R1, we can see that Fe SAC good intrinsic activity and selectivity compared to those reported materials for nitrate reduction. However, we have to say that many of the references in the table (typically in environmental engineering field) mainly focus on nitrate removal with N_2 as the target product, and do not include enough fundamental electrochemistry information for us to compare. Thus, it is might be unfair to compare the intrinsic nitrate reduction activity and selectivity of Fe SAC with most of the reported materials. To avoid misunderstanding of these performance comparisons, we have made a note like this "It might be unfair to compare the intrinsic nitrate reduction activity and selectivity of Fe SAC with most of the reported materials, because the past nitrate reduction mainly focus on nitrate removal with N_2 as the target product." in the caption of this Table.

Table R1. Comparison of performance of Fe SAC with reported catalysts by electrocatalytic nitrate reduction. Of note, it is might be unfair to compare the intrinsic nitrate reduction activity and selectivity of Fe SAC with most of the reported materials, because the past nitrate reduction mainly focus on nitrate removal with N_2 as the target product.

Catalyst	Electrolyte	NH_3 partial current	FE (%)	NO_3^- -to- NH_3 selectivity (%) ^a	Reference
Fe SAC	0.5 M KNO_3 /0.1 M K_2SO_4	98.6 $mA\ cm^{-2}$ 16324.5 $A\ g_{Fe}^{-1}$ (-0.85 V)	74.9 (-0.66V)	~69 (-0.85 V)	This work
	0.1 M KNO_3 /1.0 M KOH	60.7 $mA\ cm^{-2}$ 4019.9 $A\ g_{Fe}^{-1}$ (-0.21 V, 2 h)	86 (-0.21 V, 2 h)	81 (-0.21 V, 2 h)	
Sn	0.05 M KNO_3 + 0.1 M K_2SO_4	/	/	8	3
CL-Fe@C	100 $mg\ L^{-1}$ NO_3^- -N + 0.02 M NaCl	/	/	<2	4
Cu	0.1 M $NaNO_3$ + 0.01 M NaOH + 0.5 M NaCl	/	/	26.4	5
Co_3O_4 - TiO_2 /Ti	50 ppm NO_3^- + 0.1 M Na_2SO_4 + PVP + 1000 ppm Cl	/	/	24	6
$Pd_4Cu_4@N$ -pC	neutral NO_3^- solution	/	/	<20	7
Cu/rGO/graphite plate (GP)	0.02 M $NaNO_3$ + 0.02 M NaCl	/	/	29.9	8

Ni-Fe ⁰ @ Fe ₃ O ₄	50 ppm NO ₃ ⁻ + 10 mM NaCl	/	/	10.4	9
Pd-Cu/SS	0.01 M NaClO ₄ + 0.6 mM NaNO ₃	/	/	6	10
TiO _{2-x} /Ti foil	0.5 M Na ₂ SO ₄ + 50 ppm NO ₃ ⁻ -N	<10 mA cm ⁻² <20 A g _{Ti} ⁻¹	85	87.1	11
Pd- Cu/γAl ₂ O ₃	50 ppm NO ₃ ⁻ -N	/	/	19.6	12
Cu/Cu ₂ O NWAs	200 ppm nitrate-N + 0.5 M Na ₂ SO ₄	~100 mA cm ⁻² <200 A g _{Cu} ⁻¹	95.8	81.2	13

^a NO₃⁻-to-NH₃ selectivity = $C_{\text{NH}_3}/\Delta C_{\text{NO}_3^-} \times 100\%$, where C_{NH_3} is the concentration of NH₃(aq), and $\Delta C_{\text{NO}_3^-}$ is the concentration difference of NO₃⁻ before and after electrolysis. In main text, the selectivity mentioned refers to FE unless otherwise specified. Since Fe single atoms are active sites for Fe SAC, we also compared the metal mass activity of different catalysts. For TiO_{2-x} and Cu/Cu₂O NWAs on metal foils, the loading of catalysts is always high. We assumed a very low metal loading of 0.5 mg_{metal} cm⁻² for these catalysts on their surface, based on which we estimated NH₃ partial current per mass of metal content.

We completely agree with the reviewer that the performance of our Fe SAC seems to be not as good as the Cu/Cu₂O NWAs [Angew. Chem. Int. Ed. 2020, 59, 5350-5354] mentioned by reviewer. The performance of Cu/Cu₂O NWAs has also been included into the above table. However, we want to emphasize here that the Cu/Cu₂O NWAs is synthesized on a Cu mesh bulk electrode, and the catalyst loading should be at least several orders of magnitude higher than our catalyst loading (0.4 mg cm⁻²), not even mentioning if normalized to active metal sites. Therefore, a direct performance comparison is not reasonable. When we normalized the NH₃ partial current based on the metal loading amount by assuming a very low loading of 0.5 mg_{Cu} cm⁻² for the surface catalyst of Cu/Cu₂O NWAs (very unlikely, should be much higher loading based on their materials synthesis and characterizations), and it is found that the NH₃ mass activity of Fe SAC is two orders of magnitude higher than Cu/Cu₂O NWAs. Besides, we also tried our best to enhance performance of our Fe SAC by further optimizing our catalysis system (Figure R5). By increasing the loading of Fe SAC to 1.0 mg cm⁻² and using 0.1 M KNO₃/1.0 M KOH mixed solution as the electrolyte, the best FE can be enhanced to 81% at -0.21 V for 0.5 h electrocatalysis test. Furthermore, when we prolonged the electrocatalysis reaction time from 0.5 h to 2 h at -0.21 V, the FE can be up to 86% at 2 h test, which is comparable to that of bulk Cu/Cu₂O NWAs electrode with much higher catalyst loading. The best NO₃⁻-to-NH₃ selectivity of Fe SAC can be up to 81.0%, which is the same to that of bulk Cu/Cu₂O NWAs (selectivity of 81.2%). Meanwhile, the NH₃ mass activity of Fe SAC is one order of magnitude higher than Cu/Cu₂O NWAs. Thus now overall performance of our Fe SAC can favorably compares with bulk Cu/Cu₂O NWAs electrode, despite of much higher catalyst loading of Cu/Cu₂O NWAs electrode. The possible reason that prolonging the electrocatalysis reaction time from 0.5 h to 2 h at -0.21 V can enhance FE is described below. With the increase of electrocatalysis test time, the concentration of byproduct NO₂⁻ would increase in the electrolyte. As NO₂⁻ reduction to NH₃ was easier than NO₃⁻ reduction

(Supplementary Figure 29), thus the FE of NO_3^- -to- NH_3 would increase from 0.5 h to 2 h. Further prolongation of electrocatalysis test time would cause the decrease the concentration of NO_3^- or NO_2^- , thus HER could happen and the FE of NO_3^- -to- NH_3 would decrease. Indeed, we found that the FE of NO_3^- -to- NH_3 decreased from 86% at 2-h to 84% at 3-h test, and the NH_3 partial current density also decreased accordingly. The optimized performance of Fe SAC for nitrate reduction were also included into the Table R1 for comparison.

Figure R5. Electrocatalytic nitrate reduction performance of Fe SAC loaded on carbon paper tested in 0.1 M KNO_3 /1.0 M KOH mixed electrolyte. Catalyst loading content: 1.0 mg cm^{-2} . (a) NH_3 FE and (b) NH_3 partial current density of Fe SAC at each given potential for 0.5 h electrocatalysis test. (c) NH_3 FE, (d) NH_3 partial current density, and (e) NO_3^- -to- NH_3 selectivity of Fe SAC at -0.21 V at different time.

Finally, beside catalytic performance, we believe that developing new catalysts for nitrate reduction and understanding its catalytic mechanism are equally important. Compared to those catalysts with complicated nanostructures which are not ideal for mechanism understanding, our single atom catalyst provides a clear, well-defined active site platform and model for an in-depth understanding using DFT simulations. This can provide important guidance to future

studies in searching for better catalysts. The above Table R1 and Figure R5 have now been included into revised Supplementary Information (Supplementary Table 3 and Supplementary Figure 34).

3. It should be noted that iron has already been reported as highly selective for nitrate reduction to ammonia in 2005/2009. The efficiency is also similar to the one reported in this study.

- 10.1016/j.watres.2005.07.032

- 10.1016/j.electacta.2009.03.064

The focus of wastewater treatment is not ammonia generation, but in-fact nitrogen removal; therefore, the ammonia yield rate is not actively reported in these studies about nitrate reduction.

If the authors want to demonstrate that Fe-SAC is better than iron-based bulk metal electrodes, they must, at the least, demonstrate comparison with more than just iron nanoparticles. For example, iron mesh, foam, foil, etc.

--- We completely agree with reviewer's viewpoint that the Fe bulk have already been reported for nitrate removal, and really appreciate the reviewer providing the relevant papers to us for reference. However, Fe nanoparticle or bulk Fe have completely different materials properties compared to Fe single atom catalysts, and thus they present completely different catalytic performances. As also mentioned by Reviewer 2, our study is the first one to study single atom catalysts for nitrate reduction.

After carefully reading these two papers, we found that these two works are different from our work. Additionally, there are some problems about using bulk Fe or Fe nanoparticle electrode. So let's explain one by one. The first paper [Water Res. 2005, 39, 4065-4072] studied bulk Fe electrode for the nitrate-N transformation at pH 7 and 9 after 5 h. They found the nitrate transformation is about 80% at these two pHs after 5 h. As we already mentioned in Question 2, the nitrate transformation value is totally different from the Faraday efficiency used in our work. As they did not consider other non nitrate reduction process such as hydrogen evolution, we cannot know the Faraday efficiency of their process. More importantly, they found the Fe element would leach from bulk Fe electrode into electrolyte, which limit the use of bulk Fe electrode. What they said in their paper is like this "*The total dissolved iron concentration in the treated water was 1.67 mg/L at pH 9. Much higher values of iron than the theoretical value (<0.05 mg/l) might be due to the presence of Fe²⁺ in water. High dissolved iron concentration though not toxic for human use, its presence creates color problem of the treated water and thus can be taken as a limitation of using these electrodes*" (Page 4068 of this paper). The Fe leaching issue is also found in our FeNP/NC sample, but it is not the problem for our Fe SAC. The second paper [Electrochim. Acta 2009, 54, 4600-4606] investigated Fe electrode as cathode to electrochemical reduction of nitrate to nitrogen. The authors found that the nitrate removal was 87% and selectivity to nitrogen was 100% in 3 h with Fe cathode in the presence of NaCl. So they did not get any ammonia by using Fe electrode in their study. After all, these

two papers are focused on denitrification instead of ammonia production of our current work, and the efficiency mentioned in these two paper refers to nitrate-N transformation or nitrate removal efficiency, not ammonia faradaic efficiency used in our work.

In addition, we followed reviewer's suggestion and tested the performance of Fe foil for nitrate reduction in the same conditions as our Fe SAC (Figure R6). The NH_3 partial current density and NH_3 yield rate based on the catalyst loading amount are much lower than those of Fe SAC (Figure R6a, b), indicating the much higher catalytic activity of Fe SAC than Fe foil. In addition, Fe foil need more negative potentials to get the best FE (i.e. 71.6% at -0.77 V for Fe foil) than that of Fe SAC (~ 75% at -0.66 V, Figure R 6c), further indicating that Fe SAC possesses better nitrate reduction performance than Fe foil. The performance of Fe foil for nitrate reduction have been included into revised manuscript (Supplementary Fig. 40 in Supplementary Information).

Figure R6. (a) NH_3 partial current density and (b) NH_3 yield rate based on the catalyst loading amount of Fe SAC and Fe foil. (c) FE of NO_3^- reduction on Fe foil.

4. Moreover, in supplementary Fig. 35, they show FE of nitrate reduction to NH_3 and NO_2^- for NC and FeNP/NC. If you superimpose the subgraphs (a) on to (b), we observe barely any difference in nitrite reduction efficiency. Whereas for FeNP/NC has a curve for FE- NH_3 slightly translated in the y-axis but follows the same trend. Which begs the question: How well-controlled were these two experiments?

--- Thank you for this point. There might be some misunderstandings here. Actually, the FEs of nitrate reduction to NH_3 and NO_2^- for NC and FeNP/NC are different. To demonstrate this point, we replotted the FEs of nitrate reduction to NH_3 and NO_2^- for NC and FeNP/NC in one figure, as shown in Figure R7. We can see that the FE of nitrate reduction to NH_3 over FeNP/NC is obviously higher than that of NC at the similar potentials, and the FE of nitrate reduction to NO_2^- for NC and FeNP/NC is also different. More importantly, the partial current of ammonia on FeNP/NC is several folders higher than that on NC as shown in Fig. 3C (Page 10 in the manuscript), suggesting that these two samples are completely different.

Figure R7. FEs of NO_3^- reduction on NC and FeNP/NC catalysts at each given potential.

5. Overall this article does not bring any significant contribution to the field. Established literature has already reported better efficiencies and an understanding of the mechanism for nitrate reduction. An insight into the high yield would perhaps be of significant interest, which has not been critically analyzed.

--- We appreciate the reviewer's important suggestion and completely understand his/her concern here. We however respectfully disagree with the reviewer's point that our work does not bring any significant contribution to the field. As we have mentioned in question 2, the efficiencies reported in most of previous reports are not based on the same performance metrics in this work. The catalytic activity per active site and ammonia selectivity of our Fe SAC is among the best. Additionally, after further optimizing our catalysis system, now overall performance of our Fe SAC can favorably compares with bulk Cu/Cu₂O NWAs electrode, despite of much higher catalyst loading of Cu/Cu₂O NWAs electrode (Figure R5 and Table R1). As to the understanding of mechanism, as nitrate reduction has so many different reaction pathways towards different products, we respectfully disagree with the reviewer that the mechanism of nitrate reduction has been completely understood. Using single atom catalyst, with well-defined atomic sites for DFT simulation models, is one of the most important reasons why we studied this materials system in nitrate reduction to get a deeper reaction mechanistic understanding. We calculated each elementary step from nitrate to ammonia and figured out

the most possible pathway and intermediates on Fe single atomic site. The high yield of our catalyst can also be explained by the favored energy barriers on Fe site as calculated by our DFT simulation. In addition, as also mentioned by Reviewer 2, our work is the first one to systematically study nitrate reduction on single atom catalyst, representing a high novelty of this study. Considering the high activity of nitrate reduction on Fe single atomic site, the first case study of single atom catalysts in nitrate reduction, as well as a deep understanding in reaction mechanisms using DFT simulations, we believe that our work will make significant contributions to the field and could open up opportunities for the development of more active and selective single atom catalysts for nitrate reduction towards different products. We have now included the following discussion into our revised manuscript” (Pages 18-19). “Combining experimental results and DFT calculations, the high NH_3 yield rate or activity of Fe SAC in this study can be attributed to the following two aspects. On one hand, the Fe SAC has intrinsically high-efficiency active sites, i.e. Fe- N_4 centers, which exhibit much lower thermodynamic barriers, evidencing from smaller calculated limiting potentials, than that of Fe NP of FeNP/NC, Co- N_4 of Co SAC, and Ni- N_4 of Ni SAC. On the other hand, the optimized electrocatalytic conditions, including the concentration of KNO_3 , pH of electrolyte and applied potential, also play an important role in high NH_3 yield rate of Fe SAC.”

6. More control experiments need to be conducted to address the following:

- NC catalyst showed ammonia production Fig 3c. A control experiment to ensure that the ammonia produced is indeed nitrate reduction rather than trapped ammonia from the synthesis process (ammonium peroxydisulphate was used in synthesis).

--- Thank you for this very important point. Ammonium peroxydisulphate can completely be decomposed at 800 °C for 2-h, thus there should be no ammonia in the prepared NC. According to the reviewer’s suggestions, we carried out two independent experiments to demonstrate that the produced ammonia on NC is from nitrate reduction process, not from NC catalyst itself. In first experiment, we just soaked the NC coated glassy carbon electrode into $\text{KNO}_3/\text{K}_2\text{SO}_4$ mixed electrolyte for 0.5-h. Then we used the UV-vis to test the electrolyte before and after soaking. We found the UV-vis spectra of electrolyte before and after soaking are nearly identical (Figure R8a). In second experiment, we carried out potentiostatic test for NC coated glassy carbon electrode at ca. -0.7 V for 0.5-h in K_2SO_4 electrolyte without nitrate. The UV-vis spectra of electrolyte before and after electrocatalysis test did not change (Figure R8b). The results clearly prove that NC catalyst itself did not contribute any ammonia during its nitrate reduction process, and all of produced ammonia of NC electrode was from nitrate reduction.

Figure R8. (a) UV-vis curves of electrolytes before and after soaking NC electrode into 0.50 M KNO_3 /0.10 M K_2SO_4 mixed electrolyte for 0.5-h. (b) UV-vis curves of electrolytes before and after electrocatalysis test for NC in a K_2SO_4 solution without KNO_3 .

7. There are so many different approaches to synthesize Fe-SACs, especially those doped on a carbon support. Since SiO_2 was used as a template, were any efforts made to confirm the complete removal of Si from the prepared catalyst. Was an XPS scan for Si made during the characterization process to ensure no trace amounts of Si atoms remained on the catalyst?

--- We thank the reviewer for raising the important point here! The XPS of Fe SAC was carefully retested (Figure R9). The survey spectrum of Fe SAC is the same with our previous result in Supplementary Figure 6a. We performed zoom-in scan on the energy range from 96 to 112 eV, where Si 2p XPS spectrum locates, for 60 cycles (Figure R9b). There is no Si 2p XPS signal found, confirming the complete removal of Si from the prepared Fe SAC. The Si 2p XPS spectrum was included into revised manuscript (Supplementary Figure 6b).

Figure R9. (a) XPS survey, and (b) Si 2p spectra of Fe SAC.

8. A control experiment with a Fe-SAC on carbon support through another synthesis route without any silicon presence should be carried out to ensure there was no interference from any trace amounts of the Silicon template. (A good practice when working with single-atoms

is to minimize the no. of elements used in the synthesis process to ensure minimal contamination)

--- We appreciate reviewer's constructive suggestion here. According to the reviewer's suggestion, we prepared a Fe single atom catalyst consisting of Fe embedded in nitrogen-doped holey graphene framework (Fe-NHGF) based on a recent published Nature Catalysis paper [Nat. Catal. 2018, 1, 63-72]. Figure R10 shows the NH₃ FE of Fe-NHGF at each given potential and NH₃ yield rate and partial current density of Fe-NHGF, Fe SAC, and NC. We can see that the NH₃ FE of Fe-NHGF is very similar to that of Fe SAC presented in this work. The NH₃ yield rate of Fe-NHGF is lower than that of our Fe SAC, mainly due to the lower Fe single atom loading in Fe-NHGF compared to our Fe SAC. However, we found that the Fe-NHGF with a low Fe single atom loading still displayed a much higher NH₃ yield rate than that of NC, indicating the superior activity of Fe-N₄ active sites in Fe-NHGF. This result further confirms that the performance of Fe single atomic site is not relevant to Si.

Figure R10. (a) NH₃ FE of Fe-NHGF at each given potential. (b) NH₃ yield rate and partial current density of Fe SAC, Fe-NHGF and NC.

9. DFT Simulations. Some packages allow for simulating the solvation effect and pH effect on DFT calculations; they should be considered to provide a deeper understanding of the reaction mechanism.

--- Thanks for this kind suggestion. We used the Quantum Espresso package in which the solvation models are not implemented. Repeating all the calculations with a different code for including the solvation effect is out of the scope of this study. However, to address the reviewer's concern as mentioned in the response to the second reviewer, we included an explicit solvation model to estimate the effect of solvation on our calculated adsorption energies and potential limiting step which is critical for the conclusion of this study. As can be seen in the Figure R4 below, we have calculated the change in free energy of the HNO* formation from adsorbed NO without considering any solvation effect and explicit inclusion of a water layer. The results as shown in Table below shows that inclusion of solvation effect has a minimal effect on the free energy change associated with the potential limiting step, i.e., NO* + (H⁺ + e⁻) → HNO*. This analysis indicates that our conclusion without considering the

solvation effect is still valid. This analysis along with the figure and above discussion were added to the SI (Supplementary Figure 52).

Electrochemical Step	$\Delta G_{\text{no solvent}}$ (eV)	$\Delta G_{\text{with solvent}}$ (eV)	$U_{\text{L, no solvent}}$ (V)	$U_{\text{L, with solvent}}$ (V)
NO* \rightarrow HNO*	0.278	0.277	0.278	0.277

Figure R4. The optimized structures of (a) NO* and (b) HNO* intermediates by inclusion of explicit solvation models.

REVIEWER COMMENTS

Reviewer #1 (Remarks to the Author):

The authors addressed all my concerns. Now, the work carried out here is of high quality, I strongly recommend its publication in Nature Communications.

Reviewer #2 (Remarks to the Author):

The authors have addressed all of my comments and concerns in their manuscript revisions.

Reviewer #3 (Remarks to the Author):

The authors made commendable efforts to address earlier concerns; however, I have several reservations with regards to the following:

1. The authors should refrain from using 'highly selective' as there are more selective catalysts available. The use of 'highly' may mislead readers unless they can prove that their catalyst outperforms other researchers' average selectivity. It is currently below the reported selectivity of other catalysts, e.g., Cu, and cannot be classified as 'high'.

2. In response to comment no. 3 of review in rebuttal, the authors responded:

"The second paper [Electrochim. Acta 2009, 54, 4600-4606] investigated Fe electrode as cathode to electrochemical reduction of nitrate to nitrogen. The authors found that the nitrate removal was 87% and selectivity to nitrogen was 100% in 3 h with Fe cathode in the presence of NaCl. So they did not get any ammonia by using Fe electrode in their study."

As for no ammonia, it is because they had the presence of NaCl in the electrolyte. The paper mentions:

"On the other hand, if chloride ion was present in the solution, chlorine is generated at the anode and immediately reacts with water to form hypochlorite, which would react with ammonia during electrolysis. The overall reaction occurring in the anodic solution between hypochlorite and ammonia can be expressed as follows:

The authors suggest that Fe-SAC can have promising applications in wastewater treatment without demonstrating the proof-of-concept by treating a typical sample of wastewater containing such anions.

Based on the literature cited above, it can be inferred that Fe-SACs may not be as promising for wastewater treatment to produce NH_3 in the presence of chloride ions, rendering the recommendations of such an application baseless and misleading. Unless authors can prove otherwise, such a claim should not be present in a research article.

3. Comments on Figure R7:

- a. NC legend is confusing. It needs to be revised if part of supplementary information.
- b. Based on the graph, one can also suggest that the presence of Fe-SAC only marginally improved the performance of NC as a good nitrate-reducing catalyst to NH_3 . There remains an inherent activity of just the NC that has not been completely factored out to determine the performance of just Fe-SACs. Thus far, the evidence present is still not conclusive enough to prove Fe-SACs as an exceptional candidate for NORR.

Response to reviewers' comments

Reviewer 1

The authors addressed all my concerns. Now, the work carried out here is of high quality, I strongly recommend its publication in Nature Communications.

--- We thank the reviewer for a constructive review process as well as strong support on the publication of this work.

Reviewer 2

The authors have addressed all of my comments and concerns in their manuscript revisions.

--- We thank the reviewer for a constructive review process as well as strong support on the publication of this work.

Reviewer 3

The authors made commendable efforts to address earlier concerns; however, I have several reservations with regards to the following:

--- We thank the reviewer for a constructive review process, as well as the important suggestions which have substantially improved the quality of our manuscript.

1. The authors should refrain from using 'highly selective' as there are more selective catalysts available. The use of 'highly' may mislead readers unless they can prove that their catalyst outperforms other researchers' average selectivity. It is currently below the reported selectivity of other catalysts, e.g., Cu, and cannot be classified as 'high'.

--- Many thanks for the reviewer's very important comment here. Following the reviewer's suggestion, we used 'selective' to replace 'highly selective' in our revised manuscript.

2. In response to comment no. 3 of review in rebuttal, the authors responded: "The second paper [Electrochim. Acta 2009, 54, 4600-4606] investigated Fe electrode as cathode to electrochemical reduction of nitrate to nitrogen. The authors found that the nitrate removal was 87% and selectivity to nitrogen was 100% in 3 h with Fe cathode in the presence of NaCl. So they did not get any ammonia by using Fe electrode in their study."

As for no ammonia, it is because they had the presence of NaCl in the electrolyte. The paper mentions: "On the other hand, if chloride ion was present in the solution, chlorine is generated at the anode and immediately reacts with water to form hypochlorite, which would react with

ammonia during electrolysis. The overall reaction occurring in the anodic solution between hypochlorite and ammonia can be expressed as follows: $2\text{NH}_4^+ + 3\text{HClO} \rightarrow \text{N}_2 + 3\text{H}_2\text{O} + 5\text{H}^+ + 3\text{Cl}^-$.

The authors suggest that Fe-SAC can have promising applications in wastewater treatment without demonstrating the proof-of-concept by treating a typical sample of wastewater containing such anions. Based on the literature cited above, it can be inferred that Fe-SACs may not be as promising for wastewater treatment to produce NH_3 in the presence of chloride ions, rendering the recommendations of such an application baseless and misleading. Unless authors can prove otherwise, such a claim should not be present in a research article.

--- We appreciate the reviewer's important comment here. We guess that there should be some misunderstandings here. Due to the completely different purposes in previous work [Electrochim. Acta 2009, 54, 4600-4606] and our current work, the electrochemical cell used in previous work is very different from us. As shown in Figure R1a (**reproduced from the previous work**), the electrochemical apparatus they used is a one-chamber cell, in which anode and cathode are not separated. Thus, when there is NaCl in the electrolyte during electrolysis, hypochlorite formed on the anode side would cross to the cathode side and react with ammonia to get N_2 gas. As a result, no ammonia can be obtained. In a sharp contrast, we would like to reduce nitrate to produce ammonia, so a H-type cell was used in our work (Methods part in our manuscript, page 21). Figure R1b shows our cell for this study. You can see that our H-cell is a two-chamber cell, in which anode and cathode are separated by a cation exchange membrane, i.e. Nafion membrane. The cathode chamber was used for the nitrate reduction to ammonia in our study. If there is NaCl in the electrolyte on the cathode side, Cl^- cannot be oxidized during electrolysis, thus the produced ammonia would not be converted into N_2 gas. Even more, if there is NaCl on both anode and cathode sides, the hypochlorite (ClO^-) formed on the anode side cannot cross the Nafion membrane to cathode side to oxidize the produced ammonia, because Nafion membrane is a cation exchange membrane that cannot be crossed by ClO^- . Furthermore, we also carried out experiments to demonstrate that Fe SAC could work well for nitrate reduction to ammonia with NaCl in the electrolyte on both anode and cathode sides. As shown in Figure R1c, d, like the situation without NaCl in the electrolyte, ammonia can be produced with NaCl in the electrolyte when the potential goes to around -0.5 V. The NH_3 FE with NaCl in the electrolyte is very similar to that without NaCl (Figure R1c). The highest NH_3 FE (72.4%) with NaCl in the electrolyte is also obtained at -0.66 V, similar to that without NaCl. Besides, the Cl^- did not affect the NH_3 production rate, and the NH_3 partial current densities of Fe SAC at each given potential without and with NaCl are very close (Figure R1d). In short, Fe SAC also works well for nitrate reduction to ammonia in the presence of chloride ions. The results about nitrate reduction to ammonia on Fe SAC with NaCl in the electrolyte on both anode and cathode sides were included into revised Supplementary Information (Supplementary Figure 33).

Figure R1. (a) Schematic diagram of the electrochemical apparatus used in the mentioned paper (*Electrochim. Acta* 2009, 54, 4600-4606). (b) Digital image of H-cell used in our study. (c) NH₃ FE of Fe SAC at each given potential for 0.5 h electrocatalysis test with NaCl on both anode and cathode sides. (d) NH₃ partial current density of Fe SAC at each given potential for 0.5 h electrocatalysis tests without and with NaCl. **Of note: a is reproduced from *Electrochim. Acta* 2009, 54, 4600-4606.**

3. Comments on Figure R7:

a. NC legend is confusing. It needs to be revised if part of supplementary information.
 --- Many thanks for the reviewer's nice suggestion here. We have revised this Figure, and revised Figure is as follows (Figure R2). We also checked the supplementary information, and found that the Figures were correct.

Figure R2. FEs of NO₃⁻ reduction on NC and FeNP/NC catalysts at each given potential.

b. Based on the graph, one can also suggest that the presence of Fe-SAC only marginally improved the performance of NC as a good nitrate-reducing catalyst to NH_3 . There remains an inherent activity of just the NC that has not been completely factored out to determine the performance of just Fe-SACs. Thus far, the evidence present is still not conclusive enough to prove Fe-SACs as an exceptional candidate for NORR.

--- We appreciate the reviewer's important comment here. The NH_3 FEs for Fe SAC and NC are presented in Figure 3b (manuscript) and Supplementary Figure 38a (supplementary information). We also discussed in details in page 11 and page 14 in our manuscript. It is obvious that the NH_3 FEs of Fe SAC is higher than those of NC, which means that Fe SAC possesses a better selectivity. More importantly, the NH_3 yield rate and partial current density on Fe SAC is much higher than that on NC as shown in Figure R3 (also in Figure 3c of our manuscript), suggesting that Fe SAC possesses an much higher inherent activity than NC.

Figure R3. NH_3 yield rate and partial current density of Fe SAC and NC.